# Are Sparse Autoencoders Useful? A Case Study in Sparse Probing

**Subhash Kantamneni** [* 1]  **Joshua Engels** [* 1]  **Senthooran Rajamanoharan**  **Max Tegmark** [1]  **Neel Nanda**

## Abstract

Sparse autoencoders (SAEs) are a popular method for interpreting concepts represented in large language model (LLM) activations. However, there is a lack of evidence regarding the validity of their interpretations due to the lack of a ground truth for the concepts used by an LLM, and a growing number of works have presented problems with current SAEs. One alternative source of evidence would be demonstrating that SAEs improve performance on downstream tasks beyond existing baselines. We test this by applying SAEs to the real-world task of LLM activation probing in four regimes: data scarcity, class imbalance, label noise, and covariate shift. Due to the difficulty of detecting concepts in these challenging settings, we hypothesize that SAEs' basis of interpretable, concept-level latents should provide a useful inductive bias. However, although SAEs occasionally perform better than baselines on individual datasets, we are unable to ensemble SAEs and baselines to consistently improve over just baseline methods. Additionally, although SAEs initially appear promising for identifying spurious correlations, detecting poor dataset quality, and training multi-token probes, we are able to achieve similar results with simple non-SAE baselines as well. Though we cannot discount SAEs' utility on other tasks, our findings highlight the shortcomings of current SAEs and the need to rigorously evaluate interpretability methods on downstream tasks with strong baselines.

## 1. Introduction

Dictionary learning is a popular method for interpreting LLM activations (Arora et al., 2018; Yun et al., 2021); most

---
[*]Equal contribution  [1]Massachusetts Institute of Technology. Correspondence to: Subhash Kantamneni <subhashk@mit.edu>, Joshua Engels <jengels@mit.edu>.

*Proceedings of the 42nd International Conference on Machine Learning*, Vancouver, Canada. PMLR 267, 2025. Copyright 2025 by the author(s).

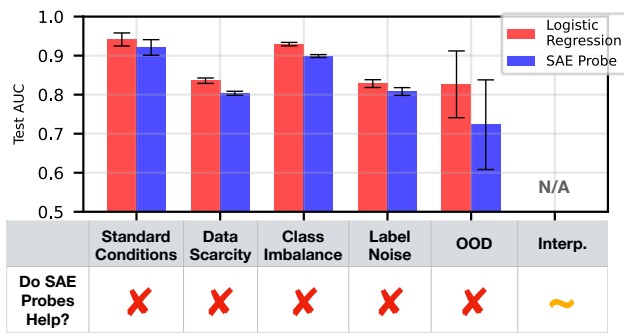

*Figure 1.* SAE probes underperform the baseline of logistic regression in each regime when taking the mean across datasets. Additionally, we find that baseline methods can provide many of the interpretability insights of SAE probes. Bars compare the best performing baseline method (logistic regression) to the best performing SAE probe ($\texttt{width} = 16\text{k}$, $k = 128$).

notably, Bricken et al. (2023); Cunningham et al. (2023) demonstrated the promise of sparse autoencoders (SAEs) and sparked considerable follow-up work. This includes papers focused on improving the downstream cross entropy loss of SAE reconstructions (Gao et al., 2024; Braun et al., 2024; Rajamanoharan et al., 2024a;b), improving SAE training efficiency (Mudide et al., 2024; Gao et al., 2024), finding weaknesses in SAEs and proposing solutions (Chanin et al., 2024; Bussmann et al., 2024b), using SAEs to understand model representation structure (Engels et al., 2024a;b; Li et al., 2024), training large suites of SAEs (Lieberum et al., 2024; He et al., 2024), and developing benchmarks for SAEs (Huang et al., 2024; Karvonen et al.; 2024b).

Unfortunately, we lack a "ground truth" to know whether SAEs truly extract the interpretable concepts used by language models. Prior work has largely avoided this limitation by evaluating SAEs with proxy metrics like reconstruction loss (Rajamanoharan et al., 2024a;b; Gao et al., 2024; Olmo et al., 2024; Braun et al., 2024). These metrics are tractable to optimize for, but do not necessarily align with mechanistic interpretability's (MI) goal of better understanding neural networks (see Bereska & Gavves (2024) and Sharkey et al. (2025) for surveys of MI). We argue that if SAEs truly advance MI's goal, they should improve performance on a real, hard-to-fake model control or explainability task.

However, despite the extensive body of work on SAEs, there are relatively few cases where SAEs have been shown to help on such a task: Marks et al. (2024) show that SAE feature circuits can help identify and remove bias from classifiers; Smith & Brinkmann (2024) show that SAEs can identify misaligned features learned by a preference model; and Karvonen et al. (2024a) show that SAE feature ablations are better at preventing regex output than baselines in a setting with scarce supervised data. While these results are promising, they all study a single example in detail, and consider comparative baselines with varying levels of rigor.

At the same time, there are also surprisingly few negative results finding that SAEs do *not* help on downstream tasks; in fact, we are only aware of Chaudhary & Geiger (2024), which finds that SAE latents are worse than neurons at disentangling geographic data, and Farrell et al. (2024), which finds that pinning related SAE latents is less effective than baselines for unlearning bioweapon knowledge. Thus, it is not clear whether SAEs are just one result away from being differentially useful, or if MI should seek fundamentally different methods. We defer a thorough examination of related work to Appendix A.

In this work, we attempt to fairly evaluate the utility of SAEs by examining their competitive advantage on a concrete task: training probes from language model activations to targets (Alain, 2016). Probing has two important qualities:

1. **Probing is practically useful:** Probing has been used to investigate LLM representations (Gurnee & Tegmark, 2023; Nanda et al., 2023; Heinzerling & Inui, 2024), detect safety relevant quantities (Zou et al., 2023), remove knowledge from models (Elazar et al., 2021), and catch synthetic sleeper agents (MacDiarmid et al., 2024).
2. **SAEs might reasonably improve probing:** Theoretically, SAE latents are a more interpretable basis of model activations, and we hypothesize that this inductive bias will help train probes in difficult regimes. Recent work has also found some positive results for the utility of SAE probes (Bricken et al., 2024a; Gallifant et al., 2025).

Thus, we curate 113 linear probing datasets from a variety of settings and train linear probes on corresponding SAE latent activations (see Figure 2). We compare to a suite of baseline methods across 4 difficult probing regimes: 1) data scarcity, 2) class imbalance, 3) label noise, and 4) co-variate shift. Unfortunately, we find that **SAE probes fail to offer a consistent overall advantage when added to a simulated practitioner's toolkit**.

Further, we explore areas that Bricken et al. (2024a) suggest SAEs may be valuable for, including detecting attributes distributed over multiple tokens and identifying dataset issues. Although SAEs initially seemed promising in these settings, we were able to achieve the same results with improved baselines. Our results underscore the necessity for

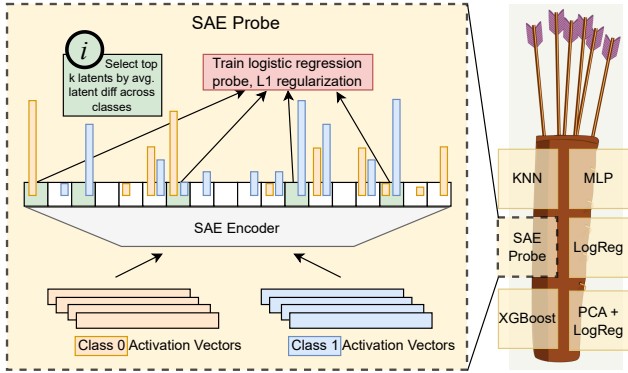

Figure 2. **Left:** An illustration of our SAE probing method. We pass in training activation vectors from each class and train an $L_1$ regularized logistic regression probe on the latents that differ the most between classes. **Right:** We ensure robustness of our results with the "quiver of arrows" approach (see Section 2.4): we add SAE regression into a set of methods, and see if the test accuracy of the best method (chosen by validation accuracy) increases.

MI works to rigorously design baselines when evaluating the utility of interpretability techniques.

## 2. Methodology

We apply probes to the hidden states of two language models, Gemma-2-9B (Riviere et al., 2024) and Llama-3.1-8B (Grattafiori et al., 2024). Our main paper results use Gemma-2-9B. We replicate core results on Llama-3.1-8B in Appendix I. We use JumpReLU (Rajamanoharan et al., 2024b) sparse autoencoders (SAEs) from Gemma Scope (Lieberum et al., 2024) for Gemma-2-9B and TopK (Gao et al., 2024) SAEs from Llama Scope (He et al., 2024) for Llama-3.1-8B. See Appendix A for further background on SAEs.

### 2.1. Classification Datasets

We collect a diverse set of 113 binary classification datasets listed in Table 4 (Appendix C) For example, 26_headline_isfrontpage requires probes to identify front-page headlines and 136_glue_mnli_entailment requires probes to identify logically entailment. For other example datasets, refer to Table 1. All datasets are titled in the form ID_description. We originally collected datasets for other purposes and discarded some of them; thus, while we have 113 datasets, ID ranges from [5, 163].

All datasets contain prompts and targets. Probes are tasked with predicting the target from the model's hidden activations when run on a prompt. We focus on binary classification, since SAEs are mostly thought of as representing binarized latents (see Bricken et al. (2023)). Thus, target is either 0 or 1. The prompts in our datasets range in length from 5 tokens to a (left-truncated) maximum of 1024 tokens.

*Table 1.* Example probing tasks. Only tasks with short prompts shown for conciseness.

| Dataset Name | Prompt | Target |
|---|---|---|
| 26_headline_isfrontpage | Sebelius's Slow-Motion Resignation From the Cabinet. | 1 |
| | MI5 References Emerge in Phone Hacking Lawsuit. | 0 |
| 36_sciq_tf | Q: Binary fission is an example of which type of production? A: asexual | 1 |
| | Q: What occurs when light interacts with our atmosphere? A: prism effect | 0 |
| 114_nyc_borough_Manhattan | INWOOD BASKETBALL COURTS | 1 |
| | PS 18 JOHN G WHITTIER | 0 |
| 149_twt_emotion_happiness | All moved in to our new apartment. So exciting | 1 |
| | is nioow calmmm eating polvoron .. yuumm | 0 |
| 155_athlete_sport_basketball | Jimmy Butler | 1 |
| | Steve Balboni | 0 |

## 2.2. Probing Strategy

To train a probe on a given dataset, we first run the model on all prompts from that dataset to generate model activations at each layer $l$ on the last token, $X_{-1}^l$. $X_{-1}^l$ is of shape $(\text{len}(\text{prompts}), \text{model\_dim})$. We probe the last token to ensure the target information was present in the preceding context. For activation probes, we then train a probe $p$ to map from $X_{-1}^l \mapsto t$, where $t$ are the targets in our dataset.

Our SAE probe training technique is summarized in Figure 2. We first pass the training dataset $X_{-1}^l$ through the SAE encoder, resulting in a batch of vectors in the SAE latent space $Z = \text{SAE}(X_{-1}^l)$ of shape $(\text{len}(\text{prompts}), W)$, where $W$ is the width of the SAE. We do not train probes directly on the SAE latent space because we hypothesize that a small number of SAE latents encodes the desired concept. Instead, we create a basis of latents with the highest average absolute difference between the set of training prompts $T_1$ with target $= 1$ and the set of training prompts $T_0$ with target $= 0$. More formally, if $\text{SAE}(X_{-1}^l) \in \mathbb{R}^W$, where $W \gg d_{\text{model}}$, we choose $k \ll W$ and find indices $\mathcal{I}$ as follows:

$$\mathcal{I} = \underset{i \in \{W\}}{\arg\,\text{top}\,\text{k}} \left| \frac{1}{|T_1|} \sum_{j \in T_1} Z_{j,i} - \frac{1}{|T_0|} \sum_{j \in T_0} Z_{j,i} \right| \quad (1)$$

We then train a probe $p_{\text{SAE}}$ to map from $Z[:, \mathcal{I}] \mapsto t$.

We note that Equation (1) has a bias toward latents with larger ranges. We thus also experiment with replacing $Z_{i,j}$ with $\frac{Z_{i,j}}{\lambda_i}$, where $\lambda_i$ is the average value of latent $i$ when it is active. This alternative selection metric leads to better results for small $k$ ($k < 32$) and similar results for larger $k$, but does not change our overall quiver of arrow results significantly (Gemma experiments use the normalized selection metric and the Llama experiments do not).

Note that previous work in SAE probing has aggregated SAE latents across tokens to provide a richer input space for

SAE probes (Karvonen et al., 2024b; Bricken et al., 2024b). However, most probing studies operate on the final token, which we choose to emulate. In Appendix K, we present results considering probes on multiple tokens.

Throughout this study, we use the area under the ROC curve (AUC) to evaluate the quality of a probe. AUC is the likelihood that the classifier assigns a higher probability of target $= 1$ to a target $= 1$ example than to a target $= 0$ example. Using AUC allows us to comprehensively assess probe performance agnostic to classification thresholds. For additional details on AUC, see Appendix D.1.

Often, a probe $p$ has hyperparameters $h_p$ we would like to optimize. We select $h_p$ that has the maximal validation AUC using the cross-validation strategy described in Table 5. We then test $p$ with optimal $h_p$ on a held out test set to calculate $\text{AUC}_p^{\text{test}}$. All datasets have at least 100 testing examples, with most having more (the average test set size is 1945).

## 2.3. Probing Methods

We use 5 baseline probing methods, detailed with their respective hyperparameters in Table 2. We evaluate 10 hyperparameter values $h_p$ for each probing method $p$. For the first three probing methods, we use grid search, while for MLPs and XGBoost, we randomly sample from a hyperparameter grid to manage the larger search space. For additional details on hyperparameter ranges, see Appendix D.3. We use all five probing methods as baselines, but for SAE probes we only use logistic regression.

## 2.4. Experimental Setup: Quiver of Arrows

To evaluate whether SAE probes provide an advantage over baselines, we use an evaluation metric we call the "Quiver of Arrows": we ask whether adding SAE probes (a new "arrow") to the set of existing probing methods available to a practitioner (the "quiver") increases performance compared to a practitioner *without* access to SAE probes. In other

| Method | Hyperparameters |
|---|---|
| Logistic Regression | $L_1$ regularization for SAE probes, $L_2$ otherwise |
| PCA Regression | Number of PCA components to reduce $X^l_{-1}$ to before using unregularized logistic regression |
| K-Nearest Neighbors (KNN) | Number of nearest neighbors |
| XGBoost (Chen & Guestrin, 2016) | n_estimators, max_depth, learning_rate, subsample, colsample_bytree, reg_alpha, reg_lambda, min_child_weight |
| Multilayer Perceptron (MLP) | Network depth, hidden state width, learning rate, and weight decay |

*Table 2.* Overview of baseline methods and their associated hyperparameters.

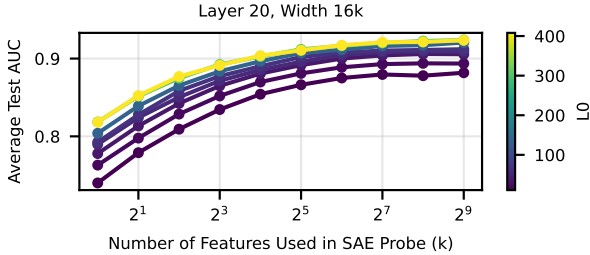

*Figure 3.* For a given width, using higher L0 and constructing probes with a larger basis of latents ($k$) is more performant.

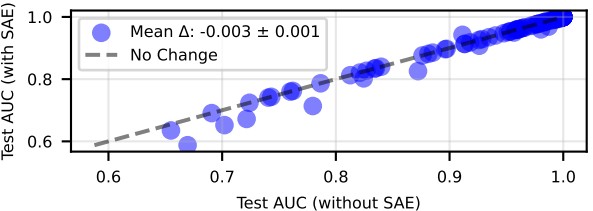

*Figure 4.* In standard conditions, when SAE probes are added to the quiver, they are chosen by validation AUC 14/133 times. However, we find a slight decrease in test AUC. Datasets not directly on the diagonal signify that an SAE method was chosen from the quiver.

words, over a collection of methods we choose the best method by validation AUC and then report the test AUC of that method. We can then compare the marginal improvement of adding SAE probes to a practitioner's toolkit by comparing the Quiver of Arrows AUC of baeline methods with and without SAE probes.

The quiver of arrows can be defined formally as follows. Given a set of probing methods $P = p_1, \ldots, p_n$ (listed in Section 2.3), we find $\text{AUC}^{\text{val}}_{p_i}$ for each method using the procedure described in Section 2.2. We then choose the probing method $p_*$ with maximal $\text{AUC}^{\text{val}}_{p_i}$ and record its $\text{AUC}^{\text{test}}_P = \text{AUC}^{\text{test}}_{p_*}$. We can then compare $\text{AUC}^{\text{test}}_P$ to $\text{AUC}^{\text{test}}_{P'}$ for a different set of methods $P' = p'_1, \ldots, p'_n$. If we let $P$ be a set of baseline methods, and $P' = P \cup \{\text{SAE probes}\}$, then $\text{AUC}^{\text{test}}_{P'} - \text{AUC}^{\text{test}}_P$ directly represents the increase in test performance when adding SAE probes to a practitioner's toolbox. We then aggregate $\text{AUC}^{\text{test}}_P - \text{AUC}^{\text{test}}_{P'}$ across many different datasets and testing regimes (e.g., dataset size) to give an overall sense of the improvement due to SAE probing.

The quiver of arrows approach is designed as a robustness check to fairly evaluate the benefit of adding SAE probes to a practioner's toolkit. If we instead constructed probes from a large set of SAEs and chose the best one based on test AUC, we could be tricked into thinking SAEs outperform baselines because we accessed the held-out test set. With the quiver of arrows approach, we emulate the information a real practitioner has access to, allowing us to make a robust

recommendation on the usefulness of SAE probes. Additional discussion of the metircs of the Quiver of Arrows is in Appendix D.4.

## 3. Comparing Probing Techniques in Different Regimes

### 3.1. Standard Conditions

We initially assess probes under standard conditions, characterized by sufficient data for probe convergence and balanced classes. We find that both baseline methods and SAE methods perform the best on layer 20 (see Figure 12a, Appendix E.1), so we run experiments with this layer. We train baselines on 1024 data points (or the maximum number of points in the dataset).

We first conduct a preliminary investigation to cut down the large space of Gemma Scope SAEs. We train probes for all SAEs using $k$ latents (for logspaced values of $k$) on all datasets using 1024 training examples. We calculate the average test AUC for each SAE probe across all datasets. We find that SAE width is relatively unimportant to probe success, while larger L0s lead to more performant probes (Appendix E.2). Additionally, as we might expect, using a larger value of $k$ leads to better probe performance, as shown in Figure 3.

Thus, throughout the paper we train probes using the largest available L0 for SAEs with width = 16k, width = 131k,

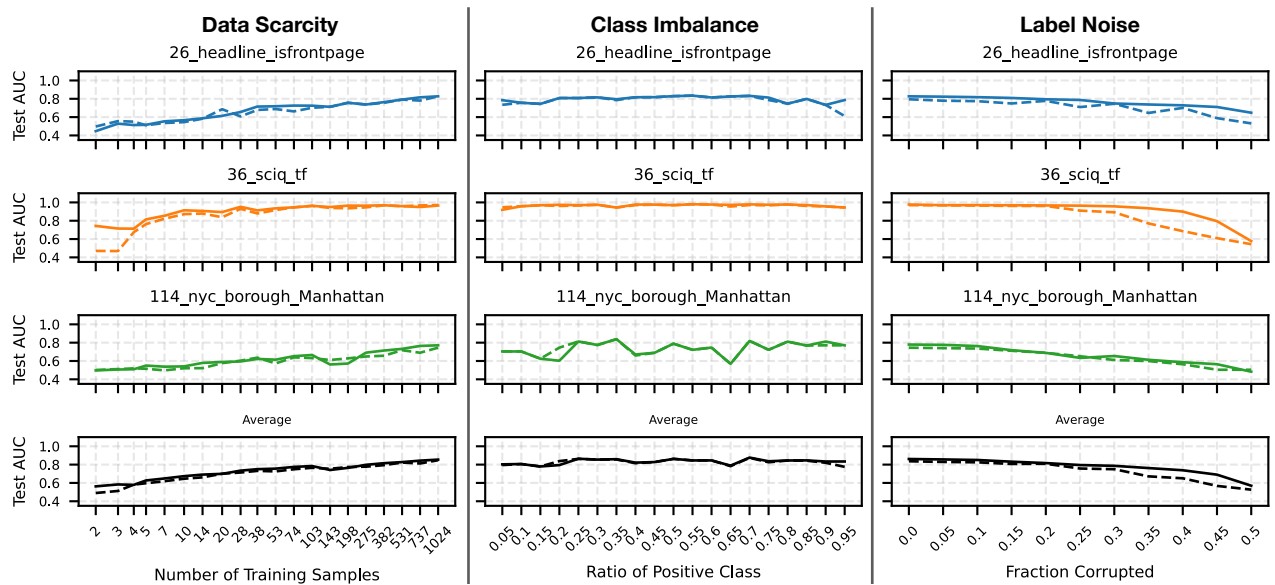

*Figure 5.* For three of the datasets in Table 1, we visualize the performance when SAE probes are in the quiver (dashed) versus when they are not (solid) for the regimes of data scarcity (left), class imbalance (middle), and label noise (right). In all three regimes, we see that on average (bottom row), SAEs do not help.

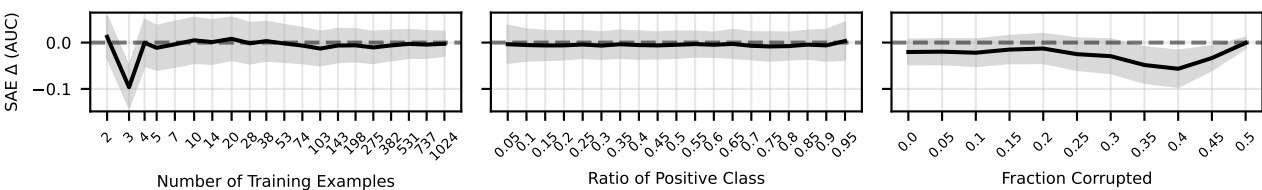

*Figure 6.* For the regimes of data scarcity, class imbalance, and label noise, we see no average improvement across datasets when adding SAEs to the quiver. Shading represents 95% confidence intervals.

and width $= 1M$. We use $k = 16$ to construct easily interpretable probes that potentially overfit less and use $k = 128$ for performance.

Using this set of probes, we consider our quiver of arrows approach in standard conditions. SAEs are chosen as the "arrow" for 14/113 tasks, however, we see a slight *decrease* in performance when they are added to the quiver in Figure 4.

It is unsurprising that SAE probes underperform baselines in standard conditions, as their inductive bias is marginal given a large, balanced dataset. Thus, we now investigate more difficult settings to test if the inductive bias of SAEs translates into a competitive advantage for probing.

### 3.2. Data Scarcity, Class Imbalance, and Label Noise

In this section, we consider more difficult probing regimes: limiting the training data (Data Scarcity), changing the relative frequency of target $= 1$ examples (Class Imbalance), and randomly flipping a fraction of the targets (Label Noise). All of these settings are realistic workflows - for

example, a researcher observing a rare model phenomena would like a probe that generalizes with few total examples or few examples of the desired class, while a researcher working with collected user data wants a generalizable probe even if some fraction of users respond randomly. See Appendix G for an explanation of why we expect SAE probes to provide a benefit in each of these regimes.

Each regime is characterized by a parameter which we vary to compare SAEs and baselines.

- Data Scarcity - We use 20 values of $n$ logspaced in $[2, 1024]$, where $n$ is the number of training examples. We use a standard test set.
- Class Imbalance - We use 19 values of ratio linearly spaced between $[0.05, 0.95]$, where ratio $= \frac{n_1}{n}$ and $n_1$ is the number of target $= 1$ training examples. The test set uses the same ratio.
- Label Noise - We use 11 values of fraction linearly spaced between $[0, 0.5]$, where fraction $= \frac{n_{\text{corrupted}}}{n}$. We use a standard test set (uncorrupted).

We evaluate the first two regimes with the quiver of arrows method. However, the quiver of arrows method relies on the validation data being representative of the test data. This assumption fails for the label noise setting since the validation data is corrupted. Thus, a hypothetical practitioner would likely deploy their most performant method from other settings instead of allowing corrupted validation AUC to arbitrarily choose a method. To emulate this, we compare logistic regression and the $\mathrm{width} = 16k$, $k = 128$ SAE probe head-to-head in the setting of label noise. We choose this SAE probe because it has the smallest width and highest $k$, which we have empirically found to be most performant.

In Figure 5, we visualize the SAE versus non-SAE quivers for a sample of datasets listed in Table 1 across the three regimes. Additionally, we visualize the average performance difference between the SAE and non-SAE quivers for each regime across all datasets in Figure 6. At all parameter values in each regime, SAEs show no meaningful improvement over baselines. This is not because SAEs are not chosen from the quiver; in Appendix F we see that SAEs are chosen for up to 40 datasets in each regime. SAEs simply underperform baselines in these settings. See Appendix F for results for individual datasets.

### 3.3. Covariate Shift

We now investigate if SAE probes are more resilient to distribution shifts in prompts. To model this covariate shift, we create or use 8 out-of-distribution (OOD) datasets: two pre-existing GLUE-X datasets which are designed as "extreme" versions of tasks 87 and 90 to test grammaticality and logical entailment respectively; three datasets (tasks 66, 67, and 73) which we alter the language of; and three datasets (tasks 5, 6, and 7) where we use syntactical alterations to names or use cartoon characters instead of historical figures. We train probes on these datasets in standard settings and evaluate on 300 covariate shifted test examples.

Like the label noise setting, our validation data is unfaithful to our test data. Thus, we compare logistic regression to a single SAE probe. We construct SAE probes from the $\mathrm{width} = 131k$, $\mathrm{L0} = 114$ SAE, as latent descriptions are available for this SAE on Neuronpedia (Lin, 2023), which facilitates later interpretability related investigations. Our results (Figure 7) show that baselines outperform SAE probes when generalizing to covariate shifted data.

## 4. Interpretability

While we find that SAE probes are not helpful in traditional probing settings, one intrinsic advantage of SAE probes is that their input basis is interpretable. To leverage this, we use the technique of automatic interpretation, or autointerp, to create natural language descriptions of top latents for

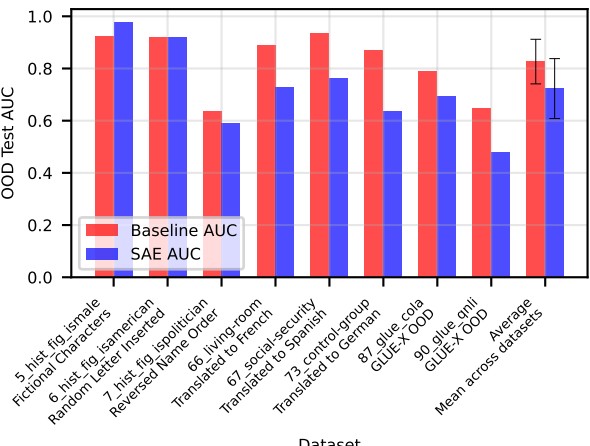

*Figure 7.* SAE probes often generalize worse than baseline logistic regression with covariate shift.

each dataset (see Bills et al. (2023)). We investigate three applications of labeled latents:

1. **Probe Interpretability**: In Section 4.1, we investigate why SAE probes fail to perform well by pruning latents that o1 (OpenAI, 2024) ranks as spurious.
2. **Latent Interpretability**: In Section 4.2, we generate autointerp explanations of each dataset's top latent to find spurious latents and latents that fit well to our probes in a way not explained by the autointerp label.
3. **Detecting Dataset Quality Issues**: In Section 4.3, we invesigate spurious dataset features and label errors using insights from the top latent descriptions and firing patterns.

### 4.1. Probe Intepretability: Pruning and Latent Generalization

We first use autointerp to investigate why SAE probes fail to generalize to covariate shift. We have two initial hypotheses: 1) the latents SAE probes rely on leverage spurious correlations in the training data and 2) the latents are not spurious, but they are not robust to the distribution shift.

First, we investigate hypothesis 1. Specifically, we attempt to improve SAE probes OOD performance by pruning latents deemed spurious by their autointerp description. We focus on three OOD tasks that SAE probes perform poorly on: 7_hist_fit_ispolitician, 66_living-room, and 90_glue_qnli (Figure 7). We use the $\mathrm{width} = 131k$, $\mathrm{L0} = 114$ SAE.

For each task, we take the $k = 8$ top latents by mean difference and use Claude-3.5-Sonnet (Anthropic, 2024) to generate latent descriptions with autointerp, in addition to one of the authors manually labeling latents. We use a smaller $k = 8$ probe in this experiment as a proof-of-concept, given the additional expense of labeling. For this and all subsequent experiments, we generate autointerp labels using

Neuronpedia (Lin, 2023), which leverages a language model to produce consistent natural language explanations for a latent based on its top activating tokens (the Neuronpedia autointerp implementation is based on (Paulo et al., 2024)). We then use OpenAI's o1 model to rank the relevance of each latent's description to the task. We also prompt o1 to downrank spurious latents given the OOD transformation. An example of this procedure for the task 66_living-room, which identifies if the phrase "living room" is in an English sentence, is shown in Table 7, Appendix J.1. For this task, o1 ranked latent 12274, which identifies "mentions of living rooms" first, while ranking latent 51330, which identifies "objects and materials related to scientific experiments," last.

We then construct a probe with the top $k$ latents using o1's relevance rank for $k \in [1, 8]$. If there are spurious latents in the $k = 8$ probe, we expect a probe with $k < 8$ to have better OOD generalization. We visualize each task's performance by $k$ in Figure 21, Appendix J.1. For two of the datasets, 66_living-room and 90_glue_qnli, we see that pruning works, with a 0.024 and 0.052 increase in AUC between the $k = 8$ and $k = 1$ probe for each dataset. However, for 7_hist_fig_ispolitician, OOD test AUC *increases* by 0.077 after pruning. This indicates that the task 7_hist_fig_ispolitician requires $k = 8$ latents to represent. This procedure is dependent on the efficacy of autointerp. To remove this confounder, a separate author ranked the relevance of human labeled latents for all three tasks, with no change to the results.

Our preliminary results show that pruning helps SAE probes generalize. However, the drop between in-distribution (ID) and OOD performance is much larger than the modest improvement from pruning. This indicates that spurious correlations are not the primary reason SAE probes fail to generalize. Thus, we consider the second hypothesis - that the underlying SAE latents do not generalize well OOD.

On the task 66_living-room, latent 122774 has an ID test AUC of 0.99 while only having an OOD test AUC of 0.64. Since the OOD transformation involves translating the prompts to French, we hypothesize that this latent is active on the phrase "living room" in English but not other languages. We use GPT-4o to translate "living room" to 15 languages, and in Figure 8 we indeed observe that latent 122774 is more active on the English translation than all other languages, and it is not active on the French translation at all. Thus, latent 122774 does not generalize OOD, which explains why the SAE probe also failed to generalize.

## 4.2. Latent Interpretability

We next generate autointerp descriptions for the most determinative latent for each dataset, allowing us to identify interesting categories of latents. Specifically, we consider each of the top 128 latents (by mean difference between

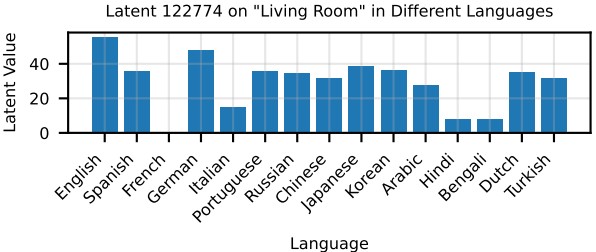

*Figure 8.* Latent 122774 is most active on the English translation of "living room," and does not fire on French.

classes) for each dataset for the $\mathrm{width} = 131\mathrm{k}, \mathrm{L0} = 114$ SAE, and evaluate each latent's $k = 1$ SAE probe in standard settings. We then generate an autointerp explanation with GPT-4o for the best latent.

Table 7 (Appendix J.2) contains a breakdown of a selection of interesting top latents that we find. We see that some latents' descriptions fit their tasks, like latent 81210 for 5_hist_fig_ismale, which activates on "references to female individuals." Another interesting category is incorrect autointerp labels (e.g. for 125_wold_country_Italy, latent 50817 has 0.989 AUC, implying (correctly) that it fires on Italian concepts; however, the description reads "names of researchers and scientific authors"). However, some latents appear to be spurious and unrelated to the semantics of their dataset. For example, 22_headline_isobama, which targets headlines published during the Obama administration, is classified with 0.782 AUC by latent 10555, which activates on "strings of numbers, often with mathematical notations."

The spurious latent category seems especially promising because finding a spurious latent may help us identify spurious features in the dataset. However, in a case study in Section 4.3.2, we find that similar findings may be possible using baseline classifiers: we apply a logistic regression probe to model hidden states on tokens from the Pile (Gao et al., 2020) and show that maximally activating examples also exhibit the spurious correlation.

However, a practical advantage for SAEs is that the infrastructure to perform autointerp is pre-existing through platforms like Neuronpedia, and a theoretical advantage is that the baseline classifier can only identify the single most relevant coarse-grained feature, while the decomposability of SAE probes into latents allows for identifying many independent features of various importance.

## 4.3. Detecting Dataset Quality Issues

In addition to generating top latent descriptions for all datasets, we present in-depth case studies on two datasets: 87_glue_cola and 110_aimade_humangpt3. These datasets were selected after an initial investigation of five datasets

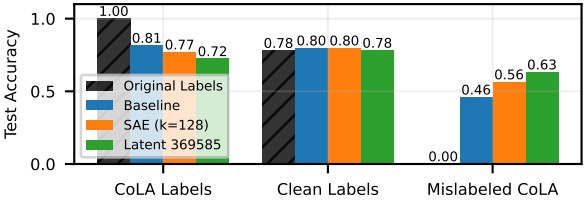

*Figure 9.* Latent 369585 outperforms a dense SAE probe and logistic regression when testing on mislabeled CoLA examples.

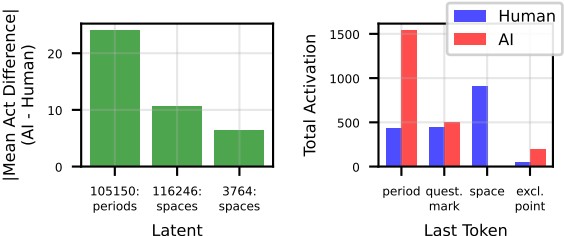

*Figure 10.* **Left:** Top 3 latents (w/ descriptions) by mean activation difference between AI generated and human SAE latent activations. **Right:** Histograms of the identity of the 4 most common last tokens in AI generated and human text.

whose top latent representations exhibited unexpectedly strong performance. During this analysis, we discovered that both 87_glue_cola and 110_aimade_humangpt3 contained labeling errors. While the SAE latents successfully highlight these errors, we observe that baseline methods are also capable of identifying them.

### 4.3.1. GLUE CoLA

We first examine 87_glue_cola, an established linguistic acceptability dataset. CoLA prompts are standard English sentences with $target = 1$ if the sentence is grammatically correct and 0 otherwise. Latent 369585 from the $width = 1M$ SAE has a test AUC of 0.76 and appears to fire on ungrammatical text. Surprisingly, when we look at prompts that latent 369585 fires on, those that it "disagrees" with the labels on (ones that are labeled grammatical), often appear to be ungrammatical. Although we first found this result with SAE probes, we later found that logistic regression was also capable of the same identification. In Table 9, we show the sentences that a baseline logistic regression classifier and latent 369585 mark as ungrammatical while the label is grammatical; most such sentences are truly ungrammatical. Thus, both SAE and baseline methods lead us to hypothesize that the CoLA dataset is partially mislabeled.

To test our hypothesis, we choose 3 LLMs - GPT-4o (OpenAI), Claude-3.5-Sonnet-New, and Llama-3.1-405B Instruct (Grattafiori et al., 2024) - to judge the linguistic acceptability of 1000 random CoLA prompts. We take the LLM majority vote as the ensembled, clean label. This is analogous to the experiment performed in Warstadt et al. (2019) that validated the CoLA dataset by ensembling the predictions of native English speakers, but with LLM judges instead of English speakers. We find that 22% of CoLA labels are mislabeled by this metric (higher than the 13% disagreement rate from Warstadt et al. (2019)). Notably, the Table 9 sentences are identified as ungrammatical by the ensemble while being labeled as grammatical in CoLA.

Next, we train a baseline logistic regression, a $k = 128$ dense SAE probe, and the latent 369585 classifier on the original CoLA labels. We then test on a held-out set with the clean, ensembled predictions. In Figure 9, we find that the original labels, the baseline classifier, the dense SAE

probe, and latent 369585 all have about the same accuracy on the clean labels. Remarkably, when we test all classifiers on examples which were misclassified in CoLA (where the ensembled labels disagree with the original labels), we find that the latent 369585 classifier outperforms the dense SAE probe, which itself outperforms the baseline considerably.

### 4.3.2. AI vs HUMAN

We also investigate 110_aimade_humangpt3, which tests if probes can distinguish between human and ChatGPT-3.5 generated text. Latent 105150 has an AUC of 0.82 on this task, yet appears to fire primarily on periods and punctuation.[1] Since this latent is syntactical and unrelated to the content of the dataset, we hypothesize it represents a spurious correlation in the dataset.

We examine the final token of each prompt in the dataset and indeed find a spurious feature: AI text is more likely to end in a period while human text is more likely to end in a space (see Figure 10). Although we discovered this by investigating SAE latents, we could reach the same conclusion with baselines: in Table 10 (Appendix J.4), we apply the logistic regression classifier to 2.8 million tokens from the Pile, and find that it is most active on punctuation tokens.

## 5. Assessing Improvements in SAE Architectures

While SAE probes do not robustly outperform baselines, a separate interesting question is whether recent SAE architectural developments have improved SAE probing performance. We investigate this question by examining the performance of eight different SAE architectures released in the last two years, as outlined in Table 3.

We use $width = 16k$ Gemma-2-2B layer 12 SAEs with a variety of L0 values trained by Karvonen et al. (2024b). We create $k = 16, 128$ SAE probes on all regimes and datasets

---

[1]See Neuronpedia for examples.

*Table 3.* Timeline of SAE Architecture Improvements.

| SAE Architecture | Publication Date |
|---|---|
| ReLU (original) | October 4, 2023 |
| Bricken et al. (2023) | |
| ReLU (updated) | April 26, 2024 |
| Conerly et al. (2024) | |
| Gated | May 1, 2024 |
| Rajamanoharan et al. (2024a) | |
| TopK | June 6, 2024 |
| Gao et al. (2024) | |
| JumpReLU | July 19, 2024 |
| Rajamanoharan et al. (2024b) | |
| BatchTopK | July 19, 2024 |
| Bussmann et al. (2024a) | |
| p-annealing | July 31, 2024 |
| Karvonen et al. | |
| Matryoshka | December 19, 2024 |
| Bussmann et al. (2024b) | |

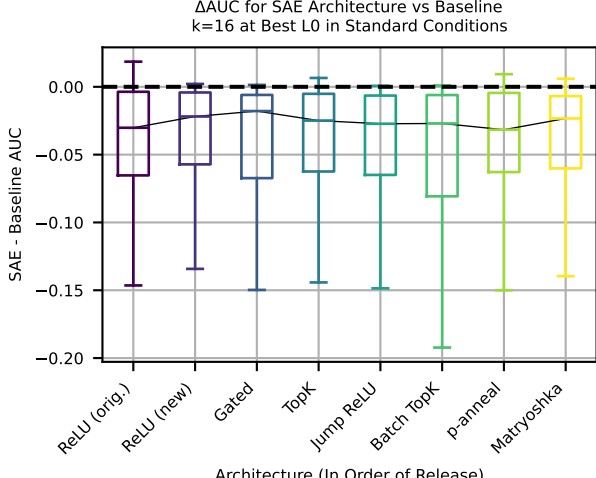

*Figure 11.* We see that there is a slight uptick in probing performance compared to baselines in standard conditions with more recent architectures. However, the spread of the data is considerably larger than this improvement.

for each architecture, and additionally $k = 1$ for standard conditions. In Figure 11, we plot each SAE architecture's $k = 16$ probe performance in standard conditions. We find the average test AUC of each SAE and then take the max across L0 for each SAE architecture (using a single L0 value is somewhat noisy, see Appendix L). This metric tests how expressive individual SAE latents are for different SAE architectures. While we see a slight positive trend for probing performance with more recently released SAE architectures, the effect is not statistically significant. For plots in all regimes, see Appendix L; in some regimes there is a slight improvement with newer SAE architectures.

## 6. Conclusion

Our evaluation of sparse autoencoders (SAEs) reveals fundamental limitations in current SAE methodologies. Despite expectations that interpretable SAE latents would provide a competitive advantage in probing, we found no improvement over traditional methods across multiple regimes and over 100 datasets. Some failures expose critical weaknesses – for instance, SAE latents struggle to be robust to distributional shifts and fail to represent complex concepts – while others are more mundane – lower L0 SAEs create worse sparse probes. More broadly, our findings highlight the need for the field of mechanistic interpretability to evaluate techniques with rigorous baselines. While prior work reported advantages for SAE probes, our robust quiver of arrows methodology, use of stronger baselines, and evaluation on a large set of datasets demonstrate these advantages to be illusory. Even in our own analysis, initial conclusions favoring SAE interpretability were later overturned when proper baselines were considered. We do not consider this work

to be a wholesale critique of the SAE paradigm. Instead, we view it as a single rigorously evaluated datapoint to contextualize SAE utility and to motivate a more thorough examination of methods in mechanistic interpretability.

**Limitations** We study the performance of SAE probes as a proxy for SAE utility. While we believe that probing performance is a more effective measurement than typical SAE evaluations like reconstruction error, it is still a proxy metric. We chose to study difficult probing regimes to try to find cases where the inductive bias of SAE latents outweighed imperfect SAE reconstruction, but it is possible that even a basis of "true" model representations would offer only a mild inductive bias advantage over traditional linear probing. We are thus excited for future work studying probing on toy models (e.g. models trained on board games (Karvonen et al., 2024c)) where the "true" model features are known.

Finally, it is possible that further optimization of the SAE probe baseline might increase performance such that it beats baseline methods. For example, Gallifant et al. (2025) find that SAE probing with a number of modifications (multi-token probes, binarization of latents, and probing with $k =$ full SAE dimension) beats baseline methods on safety-relevant datasets (albeit only comparing to single-token baseline probes). Additionally, we only tried logistic regression on SAE probes, and it is possible that other probing techniques could perform better. However, we believe a main takeaway of our paper is that equivalent effort should be put into optimizing baselines. Indeed, we examine the effect of binarizing latents in Appendix M and find that it does not significantly help.

## Acknowledgments

This work was conducted as part of the ML Alignment & Theory Scholars (MATS) Program. We would like to thank the members of the Tegmark group and Neel Nanda's MATS stream for helpful comments and feedback. JE was supported by the NSF Graduate Research Fellowship (Grant No. 2141064).

## Impact Statement

We are motivated by increasing our understanding of language models in our study of probing. There are many positive social impacts of such an increased understanding of models, including detecting deception, bias, and misalignment. We do not foresee any negative consequences of our work.

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

# A. Related Work

### A.1. Probing

Probing has a rich history in computational neuroscience, where linear decoders were used to study information representation in biological neural networks (Mur et al., 2009). This technique was later adapted to study artificial neural networks by Alain (2016). Since then, probing has become a fundamental tool in neural network interpretability, revealing that many high-level concepts are linearly represented in model activations (Gurnee & Tegmark, 2023; Heinzerling & Inui, 2024; Nanda et al., 2023). Similar to our work studying sparse probing, Gurnee et al. (2023) study sparse probing of activations and identify individual monosemantic neurons by setting $k = 1$. Recent work has also used probing to study safety-relevant properties of language models, such as truthfulness (Marks & Tegmark, 2023) and the presence of sleeper agents (MacDiarmid et al., 2024), without relying on potentially unreliable model outputs. Two recent works have investigated the utility of SAEs for probing. First, Bricken et al. (2024a) investigate a synthetic bioweapons dataset and show that SAE probes can sometimes offer an advantage against baseline probes when aggregated across multiple tokens; we discuss these results in our section on multi-token probing in Appendix K. Second, Gallifant et al. (2025) use feature binarization, multi-token feature pooling, and probing on the entire SAE vector to report that SAE probes outperform baselines. We discuss Gallifant et al.'s (2025) results in our limitations section and in Appendix M.

### A.2. Challenges in Neural Network Interpretability

Our work connects to broader concerns about the reliability of neural network interpretation methods. Adebayo et al. (2018) studied saliency methods, a classical technique for understanding models, and found that randomizing the model and dataset labels did not change many aspects of saliency maps; thus, while the method produced plausible-looking explanations, they were not faithful to the true model and dataset. Similarly, (Bolukbasi et al., 2021) demonstrated that seemingly interpretable neurons in BERT were artifacts of running on only a particular dataset rather than more general representations. For SAEs specifically, (Chanin et al., 2024) identified feature absorption and splitting as fundamental challenges, Gao et al. (2024) showed that SAEs have an irreducible error component, and in extremely recent work Heap et al. (2025) show that SAEs trained on random models also result in interpretable features. Our work extends these critiques by finding that many settings where SAE probes were thought to be helpful turn out not to be when compared to stronger baselines.

### A.3. SAE and SAE Applications

Sparse autoencoders (SAEs) provide a map from model activations to a sparse, higher dimensional latent space (Bricken et al., 2023; Cunningham et al., 2023). Individual latents are hypothesized to represent mono-semantic concepts LLMs use for computation. An SAE is parameterized by its width, or the dimension of its latent space, and its L0, or how many latents are nonzero on average. While our work focuses on evaluating SAEs as probing tools, prior work has explored various downstream applications of SAEs. Templeton et al. (2024) first used SAE latents for steering, and in follow-up work Chalnev et al. (2024) found that SAEs can help find better steering vectors (although see (Wu et al., 2025) for a very recent work finding that SAEs are not competitive for steering). Karvonen et al. (2024b) implement a set of comprehensive benchmarks for evaluating SAE performance, including SHIFT (Marks et al., 2024), sparse probing, unlearning, and feature absorption. Recent work has also used SAEs to interpret preference models (Smith & Brinkmann, 2024) and prevent unwanted behavior in model output (Karvonen et al., 2024a), both specific applications where SAEs seem to be state of the art.

# B. Disentangled Representations

An interesting connection of our work is to the field of disentangled representations, which studies representations that are *identifiable*. That is, disentangled representations have individual dimensions (or small groups of dimensions) that correspond to independent parts of the input, such that changes to other parts do not change that part of the representation (Bengio et al., 2013; Suter et al., 2019; Higgins et al., 2018). These disentangled representations are similar to sparse autoencoders latents or language model features as hypothesized by the linear representation hypothesis, and in fact prior work directly examines this connection (Joshi et al., 2025). Past work finds that disentangled representations are sometimes helpful for downstream tasks, improving sample-efficiency (Van Steenkiste et al., 2019) and generalization in classification (Lachapelle et al., 2023). Unlike these works, we find minimal benefits of plain SAE representations, perhaps implying that they are not disentangled enough, or that the probing setup w differs from these past works.

# C. Classification Datasets

Below we list in a (large) table all of the classification datasets we use, as well as their source. Note that datasets 161-163 are modified from their source -a mistake in our formatting reframes them as differentiating between news headlines and code samples.

Table 4: Binary classification tasks used.

| Citation | Dataset Name |
| --- | --- |
| Gurnee & Tegmark (2024) | 5_hist_fig_ismale |
| | 6_hist_fig_isamerican |
| | 7_hist_fig_ispolitician |
| | 21_headline_istrump |
| | 22_headline_isobama |
| | 23_headline_ischina |
| | 24_headline_isiran |
| | 26_headline_isfrontpage |
| | 114_nyc_borough_Manhattan |
| | 115_nyc_borough_Brooklyn |
| | 116_nyc_borough_Bronx |
| | 117_us_state_FL |
| | 118_us_state_CA |
| | 119_us_state_TX |
| | 120_us_timezone_Chicago |
| | 121_us_timezone_New_York |
| | 122_us_timezone_Los_Angeles |
| | 123_world_country_United_Kingdom |
| | 124_world_country_United_States |
| | 125_world_country_Italy |
| | 126_art_type_book |
| | 127_art_type_song |
| | 128_art_type_movie |
| Johannes Welbl (2017) | 36_sciq_tf |
| Lin et al. (2022) | 41_truthqa_tf |
| Ben Zhou & Roth (2019) | 42_temp_sense |
| | 130_temp_cat_Frequency |
| | 131_temp_cat_Typical_Time |
| | 132_temp_cat_Event_Ordering |
| Bisk et al. (2020) | 44_phys_tf |
| Tafjord et al. ("2019") | 47_reasoning_tf |
| Hendrycks et al. (2021) | 48_cm_correct |
| | 49_cm_isshort |
| | 50_deon_isvalid |
| | 51_just_is |
| | 52_virtue_is |
| Talmor et al. (2022) | 54_cs_tf |
| Gurnee et al. (2023) | 56_wikidatasex_or_gender |
| | 57_wikidatais_alive |
| | 58_wikidatapolitical_party |
| | 59_wikidata_occupation_isjournalist |
| | 60_wikidata_occupation_isathlete |
| | 61_wikidata_occupation_isactor |
| | 62_wikidata_occupation_ispolitician |
| | 63_wikidata_occupation_issinger |

| Citation | Dataset Name |
|---|---|
|  | 64_wikidata_occupation_isresearcher |
|  | 65_high-school |
|  | 66_living-room |
|  | 67_social-security |
|  | 68_credit-card |
|  | 69_blood-pressure |
|  | 70_prime-factors |
|  | 71_social-media |
|  | 72_gene-expression |
|  | 73_control-group |
|  | 74_magnetic-field |
|  | 75_cell-lines |
|  | 76_trial-court |
|  | 77_second-derivative |
|  | 78_north-america |
|  | 79_human-rights |
|  | 80_side-effects |
|  | 81_public-health |
|  | 82_federal-government |
|  | 83_third-party |
|  | 84_clinical-trials |
|  | 85_mental-health |
| Wang et al. (2019) | 87_glue_cola |
|  | 89_glue_mrpc |
|  | 90_glue_qnli |
|  | 91_glue_qqp |
|  | 92_glue_sst2 |
|  | 136_glue_mnli_entailment |
|  | 137_glue_mnli_neutral |
|  | 138_glue_mnli_contradiction |
| Gerami | 94_ai_gen |
| Lin et al. (2023) | 95_toxic_is |
| AI & Ishii | 96_spam_is |
| clmentbisaillon | 100_news_fake |
| Kotari | 105_click_bait |
| Davidson et al. (2017) | 106_hate_hate |
|  | 107_hate_offensive |
| Gaggar et al. (2023) | 110_aimade_humangpt3 |
| Pang & Lee (2005) | 113_movie_sent |
| AllenAI | 129_arith_mc_A |
| Rogers et al. (2020) | 133_context_type_Causality |
|  | 134_context_type_Belief_states |
|  | 135_context_type_Event_duration |
| Dublish | 139_news_class_Politics |
|  | 140_news_class_Technology |
|  | 141_news_class_Entertainment |
| Falgunipatel19 | 142_cancer_cat_Thyroid_Cancer |
|  | 143_cancer_cat_Lung_Cancer |
|  | 144_cancer_cat_Colon_Cancer |
| Kasaraneni | 145_disease_class_digestive system diseases |
|  | 146_disease_class_cardiovascular diseases |
|  | 147_disease_class_nervous system diseases |

| Citation | Dataset Name |
|---|---|
| Gupta | 148_twt_emotion_worry |
| | 149_twt_emotion_happiness |
| | 150_twt_emotion_sadness |
| Goh | 151_it_tick_HR Support |
| | 152_it_tick_Hardware |
| | 153_it_tick_Administrative rights |
| Stathead | 154_athlete_sport_football |
| | 155_athlete_sport_basketball |
| | 156_athlete_sport_baseball |
| Karvonen et al. (2024b) | 157_amazon_5star |
| | 158_code_C |
| | 159_code_Python |
| | 160_code_HTML |
| | 161_agnews_0 |
| | 162_agnews_1 |
| | 163_agnews_2 |

# D. Probing Setup

## D.1. AUC

The Area Under the Curve (AUC) quantifies the overall performance of a binary classifier using the Receiver Operating Characteristic (ROC) curve. An ROC curve plots the True Positive Rate (TPR) against the False Positive Rate (FPR) at various threshold levels, illustrating the trade-offs between correctly predicting positives and incorrectly predicting negatives. The AUC, ranging from 0 to 1, measures the entire two-dimensional area beneath this curve. An AUC of 1.0 signifies perfect classification, 0.5 indicates performance no better than random chance, and closer to 0 implies poor classification. This metric provides a single, aggregate measure of performance across all possible classification thresholds, making it particularly useful for comparing different classifiers.

Technically, when choosing $k$ and the baseline SAE to use for the other section, we use test AUC on normal datasets, which constitutes a slight leakage of test data. However, we used trends observed over a large number of datasets, in the same way we might provide heuristics for constructing SAE probes to a future practitioner.

## D.2. Probing Method Validation Details

For each strategy, we use $h_p$ that has the maximal average AUC across held out validation sets, $\text{AUC}_p^{\text{val}}$. We choose a validation method from Table 5 based on the dataset size $n$ (most of the time this is the last row in the table for large $n$, except for the low data regime tests).

*Table 5.* Selection methods to choose hyperparameters $h_p$ when training on different dataset sizes

| Data Size ($n$) | Selection method for probe $p$ |
|---|---|
| $n \leq 3$ | Train $p$ with each $h_p$ on all $n$ points; choose $h_p$ which maximizes AUC on training set. |
| $3 < n \leq 12$ | Use leave-two-out cross validation; train $p$ with each $h_p$ on all training splits of size $n-2$ and evaluate on the last 2 held out points |
| $12 < n \leq 128$ | Use 6-fold cross validation; split $n$ in sixths, train $p$ with each $h_p$ on all sets of 5 splits, and evaluate on the remaining fold |
| $n > 128$ | Use 80%/20% training/validation split |

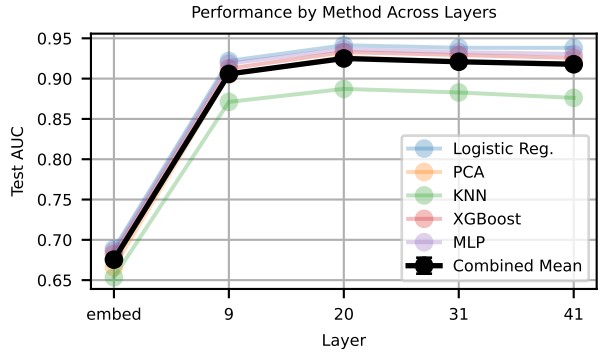
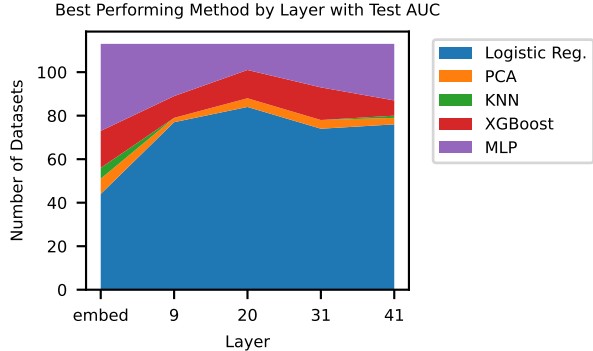

(a) Layer 20 is best for baselines.  (b) Logistic regression is often the most performant across datasets.

*Figure 12.* Analysis of layer performance and methods. (a) Shows the optimal layer for baseline performance. (b) Demonstrates the effectiveness of different methods across layers.

### D.3. Probing Method Hyperparameter Details

In this section, we list each baseline method and the hyperparameters that we search over for that method.

- **Logistic Regression**:

    - $C$: Ranges logarithmically from $10^5$ to $10^{-5}$. The $L_2/L_1$ regularization is $1/C$.

- **PCA Regression**:

    - Number of PCA Components: Varies logarithmically from 1 up to the minimum of the number of samples, latents, or 100.

- **K-Nearest Neighbors (KNN)**:

    - Number of Neighbors: Logarithmically spaced values up to the smaller of 100 or the number of samples minus one.

- **XGBoost**:

    - n_estimators: Ranges from 50 to 250 in steps of 50.
    - max_depth: Ranges from 2 to 5.
    - learning_rate: Ranges logarithmically from 0.001 to 0.1.
    - subsample and colsample_bytree: Range from 0.7 to 1.0.
    - reg_alpha and reg_lambda: Range logarithmically from 0.001 to 10.
    - min_child_weight: Ranges from 1 to 9.

- **Multilayer Perceptron (MLP)**:

    - Network depth: 1 to 3 hidden layers
    - Hidden layer width: 16,32, or 64.
    - learning_rate_init: Five values ranging logarithmically from $10^{-4}$ to $10^{-2}$.
    - alpha: Weight decay parameter, with 5 values ranging logarithmically from $10^{-5}$ to $10^{-2}$.
    - Activation function: ReLU.
    - Optimizer: Adam.

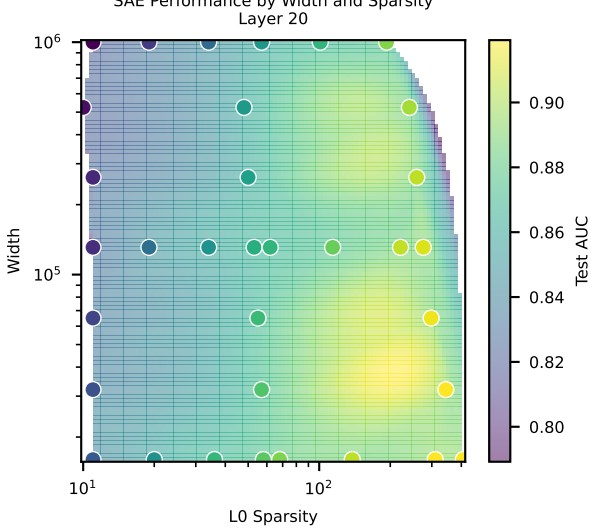

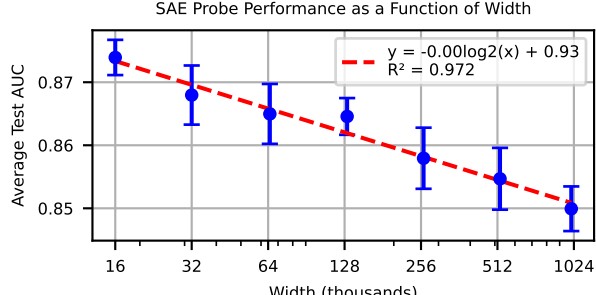

(a) We find that L0 is important in creating effective probes, while width plays a minimal role. Test AUC averaged over all values of $k$ and datasets.

(b) When averaging over other factors, smaller width SAEs perform best as probes. However, the relationship is minimal.

*Figure 14.* Analysis of SAE width and L0 regularization effects on probe performance. (a) Shows the Pareto frontier of L0 vs width trade-offs. (b) Illustrates the relationship between SAE width and probe performance.

## D.4. Additional Discussion of Quiver of Arrows

We note that adding a batch of SAE probes as additional arrows to the quiver is potentially disadvantageous for SAE probes. This is because using multiple SAE probes presents additional opportunities for these probes to overfit to the training data and be chosen by the quiver of arrows approach without properly generalizing to the test data. Nevertheless, the quiver of arrows is still appropriate because it is a proper counterfactual a practitioner would face. Additionally, the quiver of arrows approach is not the reason we find negative results for SAE probes. For instance, in Figure 1, we directly compare the test AUC of the most performant SAE probe and baseline in standard conditions across all regimes. Without using the quiver of arrows approach, we still find SAEs underperform baselines.

# E. Normal Conditions

## E.1. Cross Layer-wise Comparisons

On Gemma, we see that layer 20 is best for baselines, so we choose to use that layer moving forward (Figure 12a). Logistic regression most often has the highest test AUC across datasets in these conditions at layer 20 (Figure 12b). Finally, in Figure 13 we examine the average AUC for the width 16k SAEs with largest L0 and again find that probes at layer 20 do the best.

## E.2. Choosing which SAEs to Test

When constructing a pareto curve based on the width and L0 of all available SAEs at layer 20 of Gemma-2-9B, we find that width is unimportant, while constructing probes from a higher L0 seems to improve probe performance (Figure 14a).

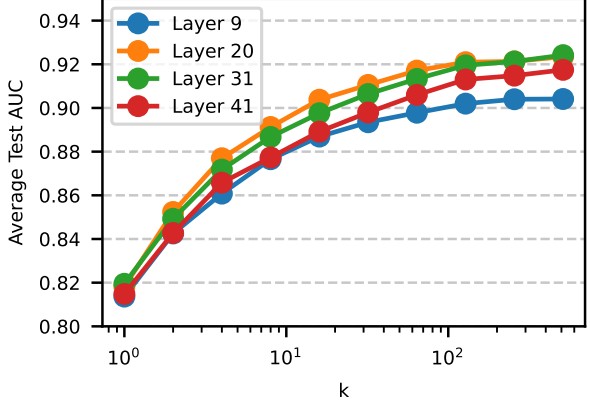

*Figure 13.* Mean test AUC, largest $\ell_0$ width 16k SAEs across layers.

To ensure that width plays a minimal role, we average over all datasets, $k$, and L0s in Figure 14b. There is a clear negative trend indicating that smaller SAE widths are better, but with almost 0 slope. Because we aim to make as few decisions based off the test data as possible, we consider three widths throughout our experiments, $16,000$, $131,000$, and $1,000,000$.

### E.3. Plotting Dataset Performance vs. K

In Figure 15, we show the performance on all datasets vs. the number of latents we train the sparse probe on. Almost all datasets are monotonically increasing in $k$; some have a sharp increase at some $k$ value, but most increase relatively smooththly.

## F. Additional Results for Various Regimes

### F.1. Data Scarcity

In the data scarce setting, we see that we choose mostly the SAE method from the quiver at smaller training fractions (Figure 17a). This is because we break ties by explicitly preferring the SAE method, and most methods have perfect validation AUC at small training fractions. Specifically, we first prefer the smallest width SAE, and then the SAE with the largest value of $k$. In Figure 16, we visualize the improvement by using the quiver of arrows approach with SAE probes across all datasets.

### F.2. Class Imbalance

We show the improvement for all datasets in Figure 16 across all class imbalances. Generally, the SAE method is chosen at more extreme imbalance ratios (Figure 17b)

### F.3. Label Corruption

We show the performance of the SAE across datasets with all values of label corruption. Interestingly, the SAE still has highest test AUC at many places (Figure 17c shows the percent of time that the SAE test AUC is larger). However, it is clear from the per dataset imshow (Figure 16) that it struggles with more label noise.

## G. Intuition for SAE Probe Utility Across Regimes

SAEs aim to construct a more interpretable basis for language model activations, with the latent space hypothesized to encode semantically meaningful and disentangled features. We argue that if SAEs are successful at this task, well-generalizing activation probes should be represented by just a few latents, and so requiring a probe to only use a sparse set of directions in this basis should serve as a beneficial inductive bias. Specifically, in setting where it is hard for dense activation probes to learn the correct direction, we argue that this inductive bias may especially help. We describe additional intuition for each regime below:

- **Data Scarcity:** Limited data makes it difficult to determine the correctly generalizing probe direction; restricting the probe to a few "true model variables" seems like a helpful inductive bias.

- **Class Imbalance:** Because SAE latents are sparsely activating, choosing SAE latents that are positive on the minority class and negative on the majority class may generalize well.

- **Label Noise:** If SAEs have a sparse basis of meaningful latents, SAE probes should be more robust to overfitting to random, noisy labels because these random labels will not be consistent with any of the latents.

- **Covariate Shift:** If SAE latents indeed use a limited set of abstract, model-internal concepts rather than dataset-specific, superficial features, then SAE probes that use a sparse set of these latents should generalize more robustly under distributional shifts in the input.

Conversely, there are a few reasons to suspect SAE probes might not be as performative as baselines. Even if SAE latents were true-to-model, abstract representations, SAEs ultimately lose information from baseline activations, leading to a disadvantage in probing. Additionally, we bound $k$, or the number of latents used in a probe, to 128, when some

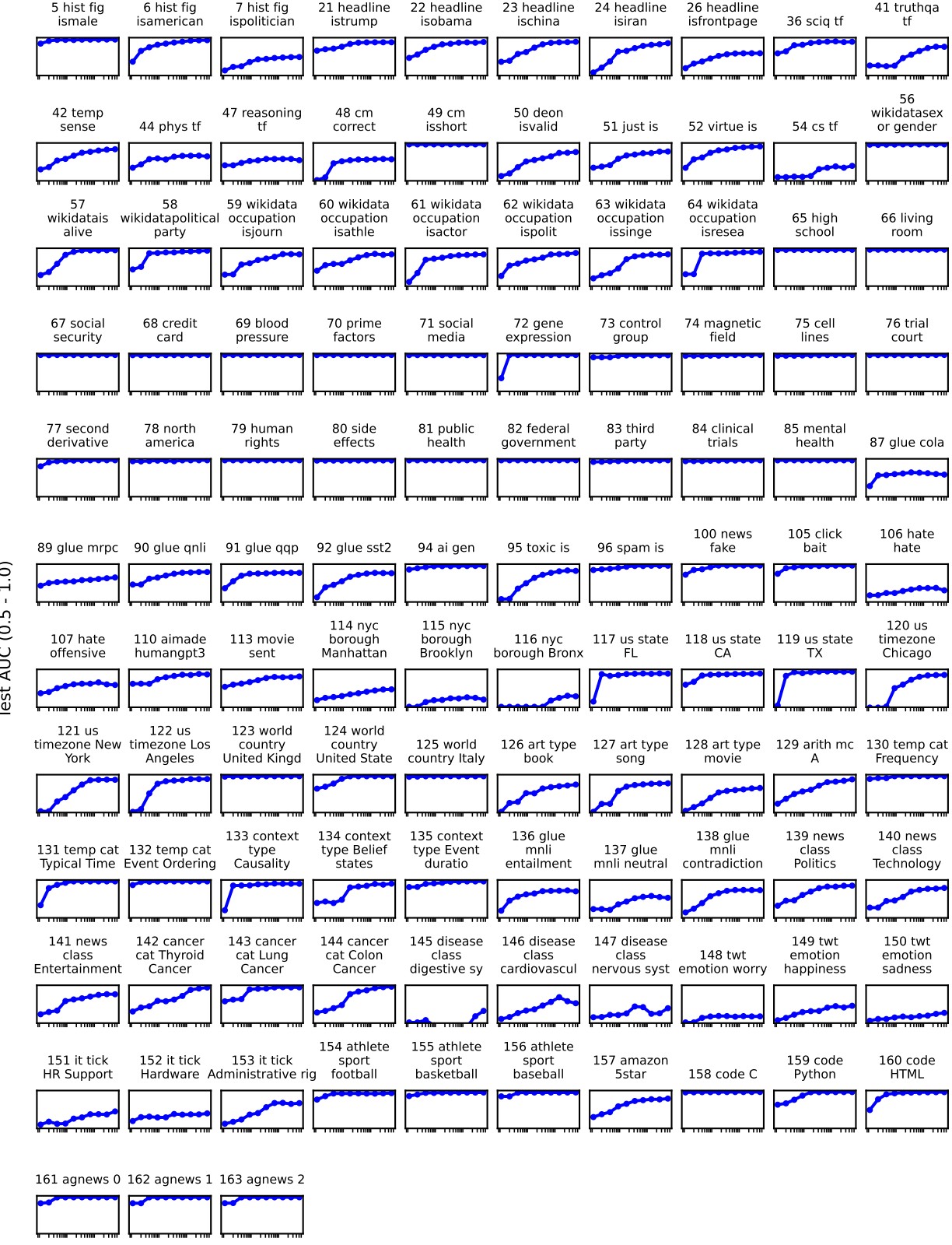

*Figure 15.* Test AUC on all datasets vs. $k = |\mathcal{I}|$ using the Gemma Scope layer 20 SAE with width 131k and $L_0 = 193$.

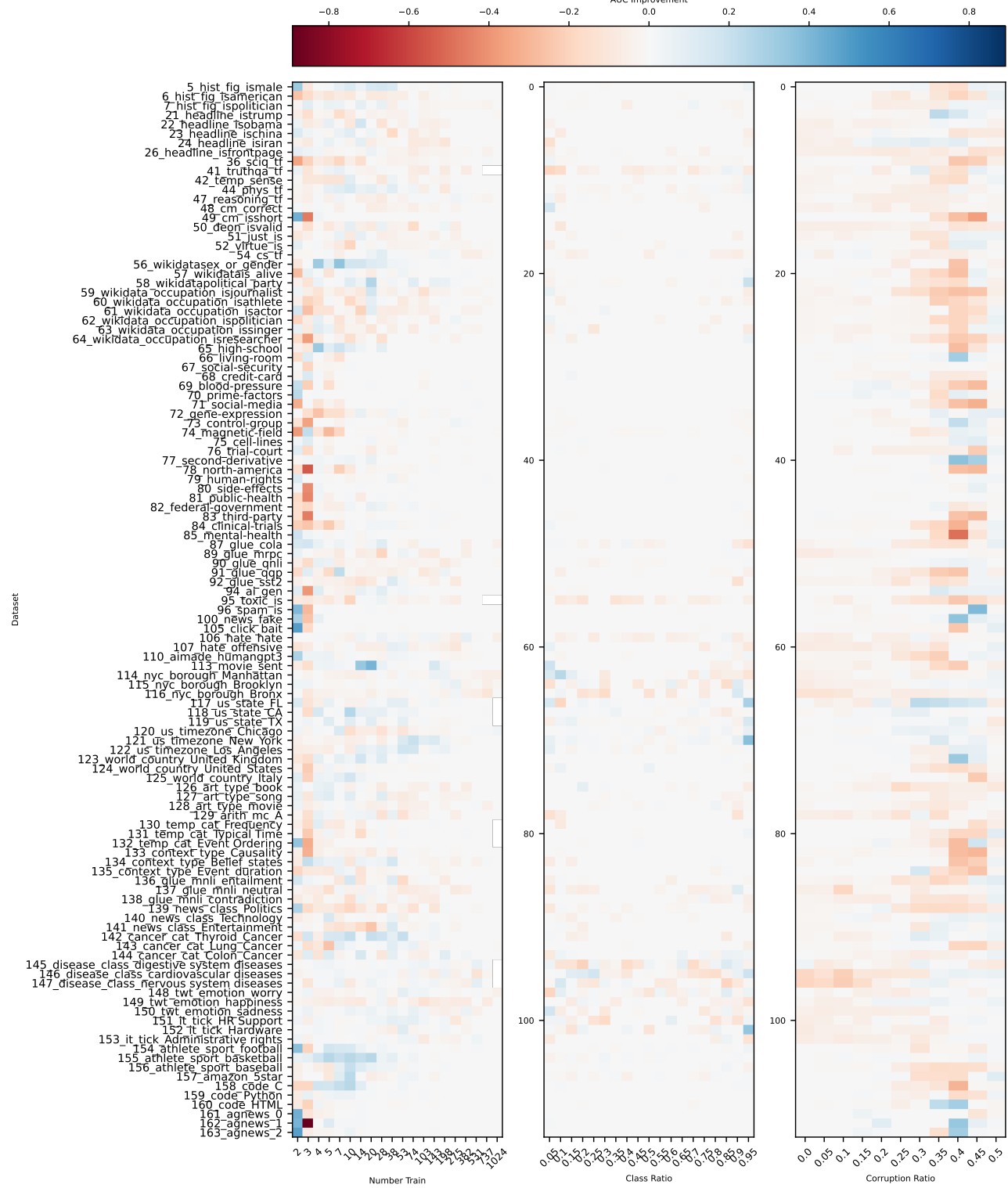

*Figure 16.* Improvement from using SAE methods in the quiver across all datasets for various corruption ratios, ratios of positive classes, and number of training examples

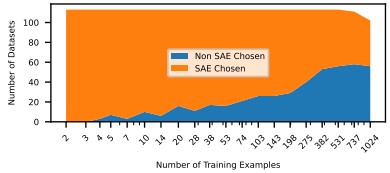

(a) SAE probes are chosen by our method at small training fractions because multiple methods have perfect validation AUC and we break ties by choosing an SAE probe.

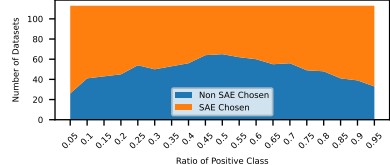

(b) SAE probes are generally chosen at more extreme imbalances.

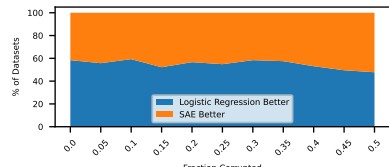

(c) Corruption fraction has a minor influence on the percent of time SAE probes or logistic regression probes are chosen.

*Figure 17.* Method selection analysis under different data conditions. (a) Selection under data scarcity. (b) Selection under class imbalance. (c) Selection under data corruption.

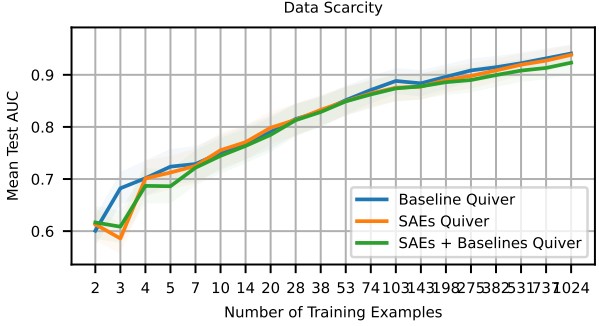
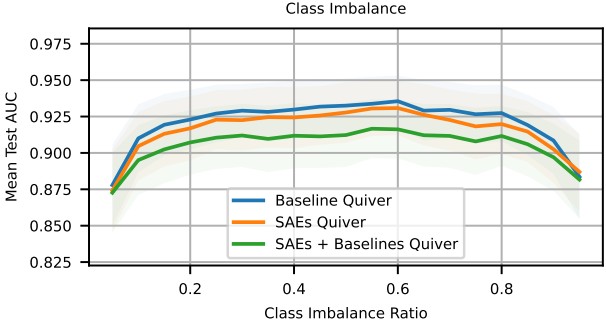

*Figure 18.* In the settings of data scarcity and class imbalance, we see that choosing an SAE probe for each task using validation AUC performs strictly worse than using the quiver of arrows with SAEs, further confirming that the quiver of arrows approach serves as an upper bound for SAE probing performance

concepts might require more than 128 latents to properly represent. Finally, it is unlikely that SAE latents are truly general, true-to-model representations, reducing their expressiveness as a probing basis.

## H. Evaluating the Validity of the Quiver of Arrows

The quiver of arrows approach we introduce is non-standard in the literature, but we adopt it to make the strongest possible case for SAEs. Since we select the best method using validation AUC, we expect to choose SAEs only for tasks where they perform best. To verify this, in the three settings where we employ the quiver of arrows—standard conditions, data scarcity, and class imbalance—we compare its performance to that of using a single SAE across all tasks, as shown in Table 6. Clearly, the quiver of arrows serves as an upper bound on the performance of any individual SAE. As an alternative counterfactual, instead of comparing against a single SAE probe, we select the best SAE for each task using validation AUC and compare this to the quiver of arrows. In Figure 18, we observe that the quiver of arrows also serves as an upper bound in the settings of data scarcity and class imbalance.

*Table 6.* We evaluate the effectiveness of the quiver by comparing it to the performance of choosing a single SAE probe for all tasks. As expected, the quiver of SAEs serves as an upper bound on the performance of any individual SAE.

| Setting | Baseline Quiver | SAEs Quiver | SAEs + Baselines Quiver | LogReg | SAE 16k k=16 | SAE 16k k=128 | SAE 131k k=16 | SAE 131k k=128 | SAE 1m k=16 | SAE 1m k=128 |
|---|---|---|---|---|---|---|---|---|---|---|
| **Standard Conditions** | 0.940 | 0.930 | 0.939 | **0.941** | 0.904 | 0.921 | 0.899 | 0.918 | 0.889 | 0.913 |
| **Data Scarcity** | 0.819 | 0.806 | 0.812 | **0.836** | 0.800 | 0.816 | 0.794 | 0.810 | 0.785 | 0.801 |
| **Class Imbalance** | 0.921 | 0.906 | 0.916 | **0.929** | 0.898 | 0.909 | 0.890 | 0.906 | 0.882 | 0.899 |

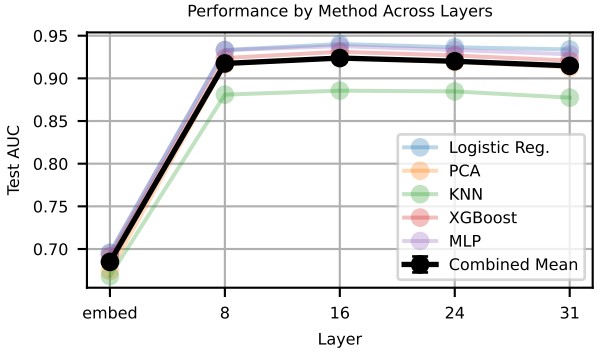
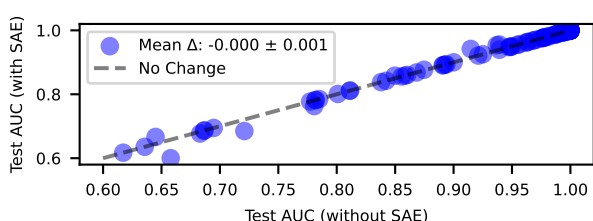

(a) When testing baseline methods at different layers of Llama-3.1-8b, we find the middle layer 16 works best by a slim margin.

(b) In standard conditions, we see that SAE probes are not statistically more performant than baselines for Llama-3.1-8b

*Figure 19.* Analysis of Llama-3.1-8b probe performance. (a) Layer-wise performance of baseline methods. (b) Comparison between SAE and baseline probes under standard conditions.

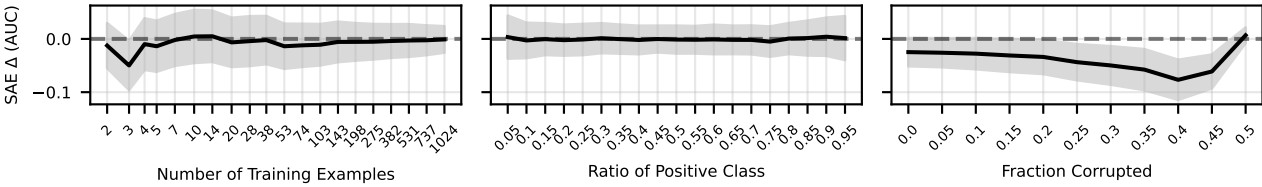

*Figure 20.* We find that Llama-3.1-8b SAE probes do not improve on baselines in the settings of data scarcity, class imbalance, and label noise.

# I. Reproducing Core Results on Llama-3.1-8b

We use SAEs provided by Llama Scope to replicate our core results, namely, that SAEs underperform baseline probes in normal, data scarce, class imbalance, and label noise settings when testing on Llama-3.1-8B. In Figure 19a, we show the results for baseline probes applied to various layers of Llama-3.1-8b in standard settings. We see that the layer roughly halfway through the model, layer 16, is best for probing experiments. In Figure 19b, we show that SAE probes in normal settings on layer 16 of Llama-3.1-8b are not more performant than baselines, although there are a few positive outliers we did not observe in Gemma-2-9b. Lastly, we use the quiver of arrows approach to show that SAE probes are not more performant in the conditions of data scarcity, class imbalance, and label noise (Figure 20). Because of the increased effort of testing OOD and interpretability, we do not replicate those results. Note that in all experiments, we use the provided width 128k, L0 55 SAE.

# J. Interpretability

## J.1. Pruning

We demonstrate the latent ranking procedure for the task 66_living-room in Table 7. We see that pruning works, for two of the tasks, 66_living-room and 90_glue_qnli, while failing for 7_hist_fig_ispolitician (Figure 21). However, even when pruning works, the improvement is marginal compared to the decrease in performance when moving from an in-distribution test set to an OOD test set. Thus, we hypothesize that the underlying latents are not sufficiently expressive. For example, latent 122774 is most active on the English translation of "living room," and not activate at all for the French translation (Figure 8). This indicates this latent does not sufficiently represent the concept of "living room" to be robust to OOD transformations.

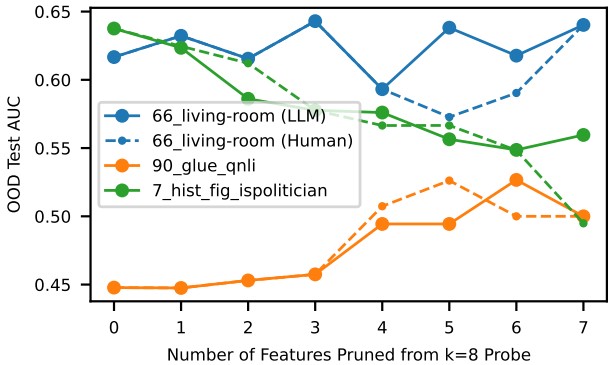

*Figure 21.* We prune using human labeler rankings (dashed) and traditional autointerp with Claude-3.5-Sonnet (solid). Two datasets improve with pruning, but one does not, which indicates that more latents are required to establish the probing target.

*Table 7.* Description of Latent Variables for Out-of-Distribution Detection. We find that this procedure ranks latents roughly according to their OOD test AUC, even though o1 did not have access to this information. However, the system is not perfect. For example, latent 51330's natural language description is ranked last and seems unrelated to the task at hand. However, its OOD test AUC is the highest of all latents. We do not investigate this further, but hypothesize that either the latent descriptions require additional tuning, or the test set truly has some spurious quality that this latent identifies.

| Latent | o1 rank | Latent Description | Val ID AUC | Test OOD AUC |
|---|---|---|---|---|
| 122774 | 1 | mentions of living rooms or parlors in houses or apartments. | 1.0 | 0.6402 |
| 98707 | 2 | phrases and words related to locations or settings, particularly indoor spaces like rooms, parlors, or living areas, as well as time references like afternoon or specific times. | 0.8698 | 0.6177 |
| 100016 | 3 | mentions of rooms or enclosed spaces, particularly when discussing physical locations or settings. | 0.6895 | 0.4736 |
| 7498 | 4 | locations within a house, particularly upstairs, downstairs, basement, bathroom, and kitchen areas. | 0.7524 | 0.6206 |
| 78823 | 5 | words and phrases related to specific places, attractions, and experiences, particularly in the context of travel and tourism. | 0.6697 | 0.4526 |
| 116246 | 6 | numbers, dates, and numerical measurements, particularly in scientific or technical contexts. | 0.7470 | 0.5398 |
| 40168 | 7 | technical terms and components related to computer systems, engineering, and scientific analysis. | 0.9219 | 0.4513 |
| 51330 | 8 | objects and materials related to scientific experiments and laboratory equipment, particularly those involving fluids, particles, or microscopic samples. | 0.7320 | 0.6483 |

## J.2. Latent Investigation

We generate natural language descriptions of all the latents with GPT-4o. We find that most are either relevant to the task or visibly spurious. We hypothesize that some latents have incomplete descriptions given their performance on the task. We list a selection of the most interesting of these latents from each category in Table 8.

## J.3. Glue CoLA

In Table 9, we show the top 5 examples where latent 369585 and the activation probe have the highest confidence that a prompt is ungrammatical, but it is (incorrectly) labeled as grammatical in the CoLA ground truth.

## J.4. AI Made

In Table 10, we show the tokens that the 110_aimade_humangpt3 classifier activates on across the pile with the highest average activation. Like the SAE feature, the baseline classifier has top activations on punctuation tokens.

# K. Multiple tokens results

Bricken et al. (2024a) report that SAE probes outperform baselines slightly. A natural question is why our experiments fail to replicate this result. One possible explanation is that Bricken et al. (2024a)'s findings were based on a single dataset, whereas we evaluate across multiple datasets. We find that SAE probes outperform baselines on only a small subset of these datasets (2.2%, see Figure 22). It is possible that the dataset used by Bricken et al. (2024a) falls within this subset; however, without access to it, we cannot confirm this.

A separate explanation involves a potential illusion due to insufficiently strong baselines. Unlike our work, Bricken et al. (2024a) uses multi-token SAE probing: they aggregate the maximum value for each latent across all prompt tokens, while we only use last token latents. Crucially, they similarly max-pool each model dimension across prompt tokens for their comparative baseline, even though activations are unlikely to have privileged dimensions (in Table 11 we show that pooling activations in this way works poorly). We implement multi-token SAE probing using max-aggregation on 60 random datasets from our list and $k = 128$; see Table 11 for full results on these datasets. We *also* implement a better baseline: we train attention-pooled probes of the form

$$\left[\text{softmax}_{t \in [1, CtxLen]}\{X_t^l \cdot q\}\right] \cdot \left[X^l \cdot v\right] \tag{2}$$

for $q, v \in \mathbb{R}^d$. We find that when compared to the last-token baseline, max-pooled SAE probes win 19.6% of the time, a considerable improvement over win rate of last-token SAE probes (2.2%, see Figure 22). However, when implementing attention-pooled baselines and using the quiver of arrows approach to select between pooled and last-token strategies for SAE probes and baselines, the SAE probe win rate drops more than 50% to 8.7% (see Figure 22).

Finally, in Table 11, we show a comparison of all methods (including the attention based probes and multi-token SAE methods discussed in Appendix K) on a random sample of 60 datasets.

# L. SAE Architectural Improvements

We test if improvements in SAE architectures have led to improvements in probing performance. We test eight SAE architectures on Gemma-2-2B in all regimes with $k = 16, 128$ SAE probes. We plot the performance of SAE probes for all architectures in Figure 24 when averaging over all datasets. While there seems to be some improvement with later architectures, we note that there is significant variance in the average across datasets that is not visualized.

# M. Assessing SAE Probe Binarization

Gallifant et al. (2025) find that binarizing latents with a threshold improves probe performance. Binarization entails setting a latent equal to 1 if its firing value is greater than a threshold, and setting it equal to 0 otherwise. Gallifant et al. (2025) use a threshold equal to 1. Gallifant et al. (2025) performs probing in the multi-token pooled setting, so to use as similar of a setup as possible we repeat our multi-token experiment with binarization and a threshold equal to 1. Following Gallifant et al. (2025), we binarize after the max-pooling aggregation. As shown in in Figure 23, we find that binarizing results in worse performance than not-binarizing.

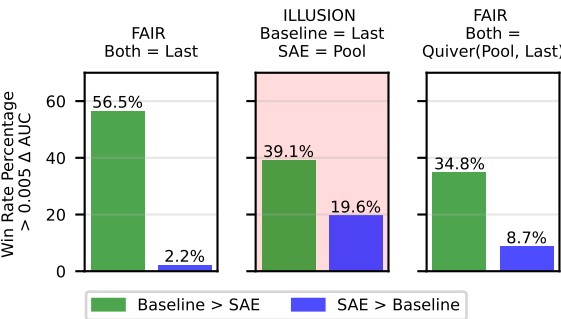

*Figure 22.* Illustration of a potentially misleading result we find: when comparing activation probes to pooled SAE probes, SAE probe win rate increases considerably, but adding in a "pooled" attention-inspired probe brings down the SAE win-rate. To clarify, the right graph compares the last token activation probe and last token SAE probe, the middle graph compares the attention softmax probe and the last token SAE probe, and the third graph compares two quivers, "select between SAE max pool probe and SAE last token probe" versus "select between activations softmax probe and activations last token probe". The win rates do not sum to 100% because we consider test AUCs within 0.005 to be tied, counting as a win for neither method.

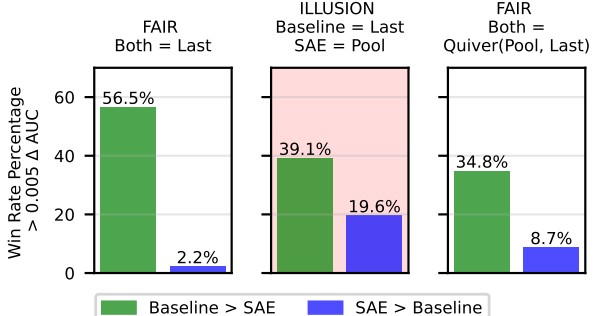

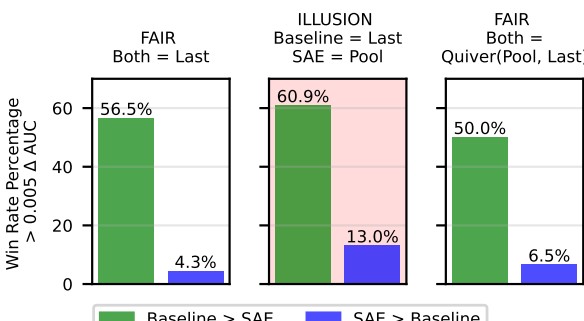

(a) Multi-token **non-binarized** experiments (the same as Figure 22).

(b) Multi-token **binzarized** experiments.

*Figure 23.* We run the multi-token experiments from Appendix K with and without binarizing aggregated latents.

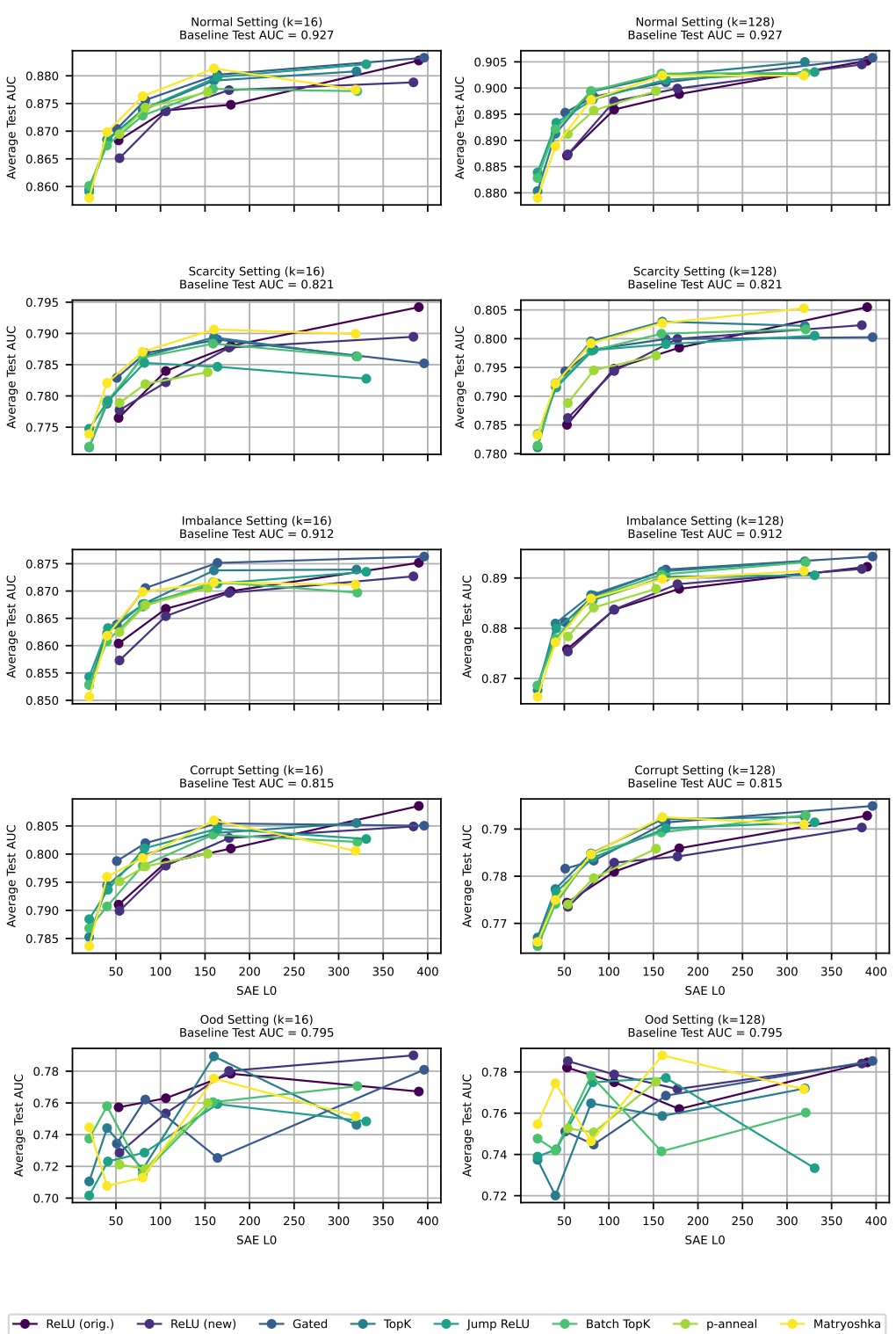

*Figure 24.* Across regimes, we see a slight positive trend in probing performance with more recent SAE architectures.

*Table 8.* Latent categorization, characteristics, and performance.

| Latent Category | Dataset | Latent Number | Test AUC | Latent Description |
|---|---|---|---|---|
| Description Fits Task | 5_hist_fig_ismale | 81210 | 0.892 | references to female individuals, especially focusing on pronouns and names. |
| | 61_wikidata_occupation_isactor | 91505 | 0.830 | names of well-known actors and film industry personalities. |
| | 66_living-room | 122774 | 0.999 | references to specific rooms in a house, particularly living rooms and parlors. |
| | 155_athlete_sport_basketball | 83267 | 0.943 | references to basketball teams, players, and related sports terms. |
| Clever Latents | 96_spam_is | 35460 | 0.907 | keywords and phrases related to text messaging and SMS communications. |
| | 100_news_fake | 31726 | 0.968 | references to conspiracy theories and secretive organizations. |
| Potentially Spurious | 21_headline_istrump | 22915 | 0.812 | numerical dates and percentages within a political or historical context. |
| | 22_headline_isobama | 10555 | 0.782 | strings of numbers, often with mathematical notations or identifiers. |
| | 110_aimade_humangpt3 | 105150 | 0.816 | statistical or numerical data and measurements. |
| | 133_context_type_Causality | 89524 | 0.947 | questions or questioning phrases, particularly those beginning with "Why?". |
| | 134_context_type_Belief_states | 113152 | 0.638 | words related to status or condition in technical or medical contexts. |
| | 135_context_type_Event_duration | 18897 | 0.876 | questions and phrases related to cost or quantity. |
| Classified by Context Length (Spurious) | 49_cm_isshort | 106376 | 1.000 | sections of text related to mathematical or programming notation and expressions. |
| | 161_agnews_0 | 106376 | 0.990 | sections of text related to mathematical or programming notation and expressions. |
| | 162_agnews_1 | 106376 | 0.988 | sections of text related to mathematical or programming notation and expressions. |
| | 163_agnews_2 | 106376 | 0.991 | sections of text related to mathematical or programming notation and expressions. |
| Task Shows Latent Description is Imperfect | 105_click_bait | 78823 | 0.967 | mentions of specific events, activities, or items in particular locations or settings. |
| | 123_world_country_United_Kingdom | 100153 | 0.978 | information related to professional roles, locations, and organizations. |
| | 125_world_country_Italy | 50817 | 0.989 | names of researchers and scientific authors. |

*Table 9.* **Left** We show the top 5 examples where latent 369585 is active while the CoLA prompt is labeled as grammatical. Most of these sentences are clearly ungrammatical, which illustrates that the CoLA data is corrupted. **Right** We then look at sentences a baseline classifier trained on CoLA marks as ungrammatical when the label is grammatical. We reach the same conclusion - the CoLA data is mislabeled.

| Prompt (Labeled Grammatical) | Latent 369585 | Prompt (Labeled Grammatical) | P(y=0) |
|---|---|---|---|
| I don't remember what all I said? | 23.09 | Rub the cloth on the baby torn. | 0.933 |
| Aphrodite said he would free the animals and free the animals he will | 15.87 | In front of them happen. | 0.926 |
| Gilgamesh wanted to seduce Ishtar, and seduce Ishtar he did. | 14.92 | Who did you give pictures of to friends of? | 0.911 |
| An example of these substances be tobacco. | 14.85 | An example of these substances be tobacco. | 0.893 |
| He will can go | 14.43 | Susan hopes herself to sleep. | 0.890 |

*Table 10.* Logistic regression classifier for 110_aimade_humangpt3's top activating tokens when run on more than 2.5 million tokens from the Pile. The classifier predominantly activates on punctuation. Only tokens with at least 10 occurences shown.

| Token | Mean Activation | Occurrences |
|---|---|---|
| \<bos\> | 6.8863 | 7625 |
| !). | 6.2529 | 10 |
| Q | 6.2271 | 1436 |
| ". | 6.0338 | 144 |
| ." | 5.9111 | 975 |
| .). | 5.7334 | 24 |
| ␣ | 5.5035 | 17 |
| ." | 5.4455 | 1057 |
| ". | 5.4132 | 319 |
| }$. | 5.3990 | 24 |

*Table 11.* Comparison of different logistic regression ("base") and SAE probing methods on a selection of 60 random datasets. Last is last token probing as in the rest of our paper, concat is concatenating the top 20 PCA dimensions of all tokens, mean is taking the mean across activation dimensions across the context, max is taking the max across activation dimensions across the context, and Attn-like probe is described in Appendix K.

| Dataset | Base (last) | Base (concat) | SAE (last) l0=68 | SAE (mean) l0=68 | SAE (max) l0=68 | SAE (last) l0=408 | SAE (mean) l0=408 | SAE (max) l0=408 | Attn -Like Probe |
|---|---|---|---|---|---|---|---|---|---|
| 6_hist_fig | **0.990** | 0.977 | 0.982 | 0.914 | 0.984 | 0.983 | 0.883 | 0.985 | 0.987 |
| 7_hist_fig | 0.750 | 0.698 | 0.738 | 0.661 | 0.739 | 0.746 | 0.613 | 0.738 | **0.760** |
| 21_headlin | 0.987 | 0.986 | 0.950 | 0.993 | **1.000** | 0.964 | 0.990 | 1.000 | 1.000 |
| 24_headlin | 0.991 | 0.979 | 0.892 | 0.995 | 0.997 | 0.947 | 0.994 | **0.998** | 0.996 |
| 44_phys_tf | 0.885 | 0.541 | 0.836 | 0.657 | 0.760 | 0.854 | 0.656 | 0.790 | **0.903** |
| 48_cm_corr | 0.833 | 0.659 | 0.788 | 0.699 | 0.779 | 0.794 | 0.708 | 0.784 | **0.860** |
| 54_cs_tf | 0.695 | 0.554 | 0.689 | 0.588 | 0.680 | **0.708** | 0.580 | 0.697 | 0.697 |
| 59_wikidat | **0.961** | 0.787 | 0.933 | 0.809 | 0.864 | 0.939 | 0.809 | 0.875 | 0.935 |
| 62_wikidat | **0.981** | 0.878 | 0.948 | 0.892 | 0.923 | 0.959 | 0.898 | 0.925 | 0.954 |
| 63_wikidat | **0.959** | 0.840 | 0.925 | 0.869 | 0.893 | 0.946 | 0.848 | 0.862 | 0.928 |
| 66_living- | **1.000** | 0.985 | 1.000 | 0.968 | 0.999 | 1.000 | 0.950 | 0.998 | 0.998 |
| 67_social- | **1.000** | 0.992 | 0.999 | 0.985 | 0.999 | 1.000 | 0.971 | 0.998 | 0.999 |
| 68_credit- | **1.000** | 0.959 | 0.998 | 0.942 | 0.986 | 0.998 | 0.920 | 0.981 | 0.993 |
| 71_social- | 0.998 | 0.985 | **0.998** | 0.969 | 0.993 | 0.998 | 0.950 | 0.989 | 0.995 |
| 73_control | **0.998** | 0.970 | 0.995 | 0.945 | 0.984 | 0.998 | 0.935 | 0.986 | 0.982 |
| 78_north-a | **1.000** | 0.933 | 0.998 | 0.907 | 0.990 | 0.999 | 0.883 | 0.976 | 0.995 |
| 79_human-r | **0.999** | 0.974 | 0.995 | 0.964 | 0.988 | 0.996 | 0.947 | 0.990 | 0.992 |
| 89_glue_mr | **0.891** | 0.761 | 0.785 | 0.784 | 0.796 | 0.823 | 0.796 | 0.777 | 0.867 |
| 90_glue_qn | 0.926 | 0.729 | 0.886 | 0.830 | 0.909 | 0.910 | 0.859 | 0.909 | **0.931** |
| 94_ai_gen | **0.998** | 0.996 | 0.993 | 0.997 | 0.996 | 0.996 | 0.996 | 0.997 | 0.996 |
| 96_spam_is | **0.999** | 0.999 | 0.997 | 0.996 | 0.997 | 0.998 | 0.995 | 0.999 | 0.999 |
| 100_news_f | **1.000** | **1.000** | 1.000 | 1.000 | **1.000** | 1.000 | 1.000 | **1.000** | 1.000 |
| 105_click_ | 1.000 | **1.000** | 1.000 | 0.999 | 1.000 | 0.999 | 1.000 | 1.000 | 1.000 |
| 106_hate_h | 0.722 | 0.612 | 0.661 | 0.742 | 0.785 | 0.674 | 0.735 | 0.788 | **0.812** |
| 110_aimade | 0.975 | 0.865 | 0.953 | 0.919 | 0.970 | 0.953 | 0.916 | 0.960 | **0.979** |
| 114_nyc_bo | 0.780 | 0.755 | 0.740 | 0.716 | 0.797 | 0.745 | 0.733 | 0.805 | **0.872** |
| 119_us_sta | **0.997** | 0.957 | 0.994 | 0.982 | 0.990 | 0.994 | 0.952 | 0.992 | 0.995 |
| 120_us_tim | 0.944 | 0.803 | 0.942 | 0.785 | **0.946** | 0.936 | 0.751 | 0.940 | 0.941 |
| 121_us_tim | 0.952 | 0.815 | 0.948 | 0.776 | 0.954 | 0.950 | 0.754 | **0.957** | 0.951 |
| 124_world_ | 0.999 | 0.999 | **0.999** | 0.965 | 0.999 | 0.998 | 0.968 | 0.999 | 0.999 |
| 127_art_ty | 0.913 | 0.872 | 0.894 | 0.834 | 0.889 | 0.905 | 0.845 | 0.900 | **0.916** |
| 129_arith_ | **0.979** | 0.903 | 0.880 | 0.867 | 0.951 | 0.938 | 0.861 | 0.975 | 0.977 |
| 132_temp_c | **1.000** | **1.000** | **1.000** | **1.000** | **1.000** | **1.000** | **1.000** | **1.000** | 0.991 |
| 133_contex | 0.984 | 0.982 | 0.975 | 0.973 | 0.979 | 0.968 | 0.973 | 0.980 | **0.986** |
| 134_contex | 0.985 | 0.977 | 0.953 | 0.966 | 0.989 | 0.966 | 0.963 | **0.994** | 0.987 |
| 137_glue_m | **0.856** | 0.551 | 0.788 | 0.632 | 0.734 | 0.807 | 0.658 | 0.742 | 0.844 |
| 139_news_c | **0.978** | 0.950 | 0.944 | 0.972 | 0.953 | 0.962 | 0.967 | 0.973 | 0.965 |
| 141_news_c | 0.951 | 0.942 | 0.917 | 0.967 | 0.968 | 0.933 | 0.965 | **0.973** | 0.965 |
| 144_cancer | **1.000** | 1.000 | 0.981 | **1.000** | **1.000** | 0.991 | **1.000** | **1.000** | 1.000 |
| 145_diseas | 0.618 | 0.677 | 0.501 | 0.618 | 0.678 | 0.563 | 0.606 | **0.686** | 0.655 |
| 149_twt_em | 0.787 | 0.756 | 0.730 | 0.780 | 0.843 | 0.759 | 0.813 | **0.853** | 0.851 |
| 150_twt_em | 0.710 | 0.612 | 0.652 | 0.700 | 0.730 | 0.672 | 0.717 | 0.746 | **0.777** |
| 155_athlet | 0.995 | 0.988 | 0.990 | 0.970 | 0.988 | 0.988 | 0.960 | 0.985 | **0.996** |
| 158_code_C | **1.000** | 0.998 | 1.000 | 0.999 | 1.000 | 1.000 | 0.999 | 1.000 | 0.997 |
| 162_agnews | **1.000** | **1.000** | **1.000** | 1.000 | **1.000** | **1.000** | **1.000** | **1.000** | 1.000 |
| 163_agnews | **1.000** | 1.000 | 1.000 | 1.000 | **1.000** | **1.000** | **1.000** | **1.000** | 1.000 |

