# OpenReview forum: "Are Sparse Autoencoders Useful? A Case Study in Sparse Probing"
_ICML.cc/2025/Conference — ICML 2025 poster_

### Official Review · Reviewer_pVWC · 2025-03-04

**Overall Recommendation:** 4

**Summary:**

This paper studies the benefits and limits of Sparse Autoencoders (SAEs). The topic is quite relevant given they are attracting more and more research in Large Language models, specifically in the context of Mechanistic Interpretability (MI). The authors propose a set of benchmarks to study both probing and interpretability of SAEs, concluding that they often do not meet their promises.

**Claims And Evidence:**

Being an experimental verification of the advantage of SAEs, claims are about the investigation. The authors cover different settings, including 5 settings for probing, and 3 settings for interpretability of SAEs.

The claims are clear, and evidence and counter-evidence have been well studied.
I have a few remarks on the interpretability side, see next box.

**Essential References Not Discussed:**

N/A

**Experimental Designs Or Analyses:**

The experimental setup seems sound and the analysis in Section 3 is accurate. Section 4 is less convincing. Section 5 is very clear.
Also, **the code is not available and reproducibility cannot be verified**.

**Methods And Evaluation Criteria:**

Linear probing seems convincing (section 3). After obtaining the last token embeddings, the investigation resembles a standard machine learning pipeline. The evaluation criteria are good. The authors' methodology is precise and both plots and settings are clear and easy to understand. Also connected, section 5 contributes to highlighting SAEs are not useful for downstream tasks.


The investigation on the SAEs activations interpretability is less rigorous compared to the previous one. And whether SAEs extract interpretable concepts remains essentially unsolved.


0. The authors start from the assumption that, somewhat, activations of SAEs are interpretable (Point 2, lines 87-93; First paragraph, lines 300-303). I am not convinced this is the case as the authors also highlight that "we lack a ground-truth to know whether SAEs truly extract interpretable concepts..." (lines 40-44). The purpose of Section 4 is to investigate interpretability by measuring usefulness for other tasks, but this does not seem to provide clear evidence in support of or against interpretability of SAEs.

1. The details about the `autointerpr` method are missing and they should be discussed at least in the supplementary. I am not even convinced that using this method is the best practice: it looks a bit circular to study the LLM concepts with another LLM, and in the best possible scenario that would have been done with a user study. In the discussion of Section 4.1, the authors focus on the latent 122774 of the SAE, whose supposed meaning is "mentions living room". This "concept", however, is not invariant to language variations (the French example), so is it really encoding the concept of "presence of living room"? Is it activating also on unrelated sentences?

2. In section 4.2, the authors further suggest that some latents are correctly labelled and some others are mislabelled by `autointerpr`. This conclusion is drawn by restricting to the dataset of interest for these latents. E.g., for positive evidence of 81210 on the dataset 5, for negative evidence of 50817 on the dataset 125. Despite that, is it the case that latents are disentangled from other potentially unrelated concepts? Can they activate in other tasks as well? Also, from the conclusion in lines 372-384 it is not clear what the authors mean.

**Other Comments Or Suggestions:**

Repetition of reference Bricken 2024a/b.

**Other Strengths And Weaknesses:**

It is sidelined that SAEs are not truly intrinsically interpretable. This has to be verified somehow.

**Questions For Authors:**

All those appearing in Methods and Evaluation Criteria. All questions revolve around the interpretability of SAEs. The main question is: Are SAE neurons interpretable?

**Relation To Broader Scientific Literature:**

Some works are worth mentioning. I would suggest the authors to connect to other representation learning works relevant to study interpretability and advantages of probing with SAEs. There is an interesting link to identifiability and disentanglement research:

[1] Are Disentangled Representations Helpful for Abstract Visual Reasoning?,  van Steenkiste et al., NeurIPS 2019 - discusses whether disentangled (which means sometimes interpretable) representations are helpful for tasks, concluding they are not. \
[2] Synergies between Disentanglement and Sparsity: Generalization and Identifiability in Multi-Task Learning, Lachapelle et al., ICML 2023 - considers sparsity and label classification to extract disentangled representations. This seems to aid in classification over new tasks. \
[3] Identifiable Steering via Sparse Autoencoding of Multi-Concept Shifts, Joshi et al., arXiv 2025 - studies identifiability of SAEs and connects to the linear representation hypothesis.

**Theoretical Claims:**

N/A

---

> ### Author Rebuttal · Authors · 2025-03-31
>
> Thank you for taking the time to review our work! We are grateful for your time and help, especially related to missing discussion of related work and questions about the automated interpretability techniques we use. We were especially glad to hear that you appreciated the depth to which we studied evidence and counter-evidence.
>
>
> ---
>
>
> >And whether SAEs extract interpretable concepts remains essentially unsolved.
>
> >.. but this does not seem to provide clear evidence in support of or against interpretability of SAEs…All questions revolve around the interpretability of SAEs. The main question is: Are SAE neurons interpretable?
>
> This is an important point that we would like to clarify. In this paper, we do not directly investigate whether SAEs discover an interpretable basis of latents. This statement is difficult to falsify or prove because there is no ground-truth for SAE latents, as you note. Instead, the goal of our work is to attempt to evaluate how “good” the basis of features that SAEs discover is by  studying how helpful they are on the downstream task of probing. This question is both more useful for practitioners and easier to answer than pure interpretability metrics. Thus, our work also provides a more objective (if indirect) measure of SAE latent interpretability.
>
> > The details about the autointerpr method are missing and they should be discussed at least in the supplementary. I am not even convinced that using this method is the best practice: it looks a bit circular to study the LLM concepts with another LLM, and in the best possible scenario that would have been done with a user study.
>
> Thank you for bringing this up! We have added the following description of autointerp in Section 4.1: “For this and all subsequent experiments, we generate autointerp labels using Neuronpedia, which leverages a language model to produce natural language explanations for a latent based on its top activating tokens (the Neuronpedia autointerp implementation is based on [4]).” Autointerp is a standard procedure for evaluating SAE latents [1, 2, 3, 4 ], although you are correct in stating that it is somewhat circular to study an LLM with an LLM! More specifically, our procedure is dependent on the efficacy of autointerp. To remove this confounder in Section 4.1 (about probe pruning), we ran an additional experiment where two of the authors independently labeled latents and ranked their relevance for all three tasks, with no change to the results.
>
> [1] "Language models can explain neurons in language models."
>
> [2] "Towards Monosemanticity: Decomposing Language Models With Dictionary Learning"
>
> [3] "Scaling and evaluating sparse autoencoders."
>
> [4] "Automatically interpreting millions of features in large language models."
>
> > In the discussion of Section 4.1, the authors focus on the latent 122774 of the SAE, whose supposed meaning is "mentions living room". This "concept", however, is not invariant to language variations (the French example), so is it really encoding the concept of "presence of living room"? Is it activating also on unrelated sentences?
>
> This is correct! We use latent 122774 as an example of a latent that doesn’t represent the true living-room feature (which we expect to be the same across languages). This indicates that the underlying SAE is imperfect, and is a possible reason why SAE probes did not generalize as well to covariate shifts. This latent generally activates only when living-room is in the sentence, but other latents are less precise.
>
> > In section 4.2, the authors further suggest that some latents are correctly labelled and some others are mislabelled by autointerpr… Despite that, is it the case that latents are disentangled from other potentially unrelated concepts? Can they activate in other tasks as well? Also, from the conclusion in lines 372-384 it is not clear what the authors mean.
>
> Thank you for mentioning this! If SAEs worked perfectly, then we would expect latents to specialize and only fire on a single concept (which could be quite complex itself). However, since SAEs are imperfect, many latents remain polysemantic (are active on multiple concepts). We have adjusted the language of lines 372-384, we agree that it was complex! Please see our response to reviewer 3 (gmH6), which has the revised paragraphs.
>
> >Also, the code is not available and reproducibility cannot be verified.
>
> Thank you for pointing this out! We have uploaded our code anonymously here: https://anonymous.4open.science/r/SAE-Probes-B404 We also have added a link to the de-anonymized github repo in the non-anonymous version of our paper.
>
> Thank you for providing the additional citations, we will add them to our related work.
>
> ---
>
>
> Thank you again for taking the time to review the paper and providing helpful feedback! Do the above actions address your concerns with the paper? And are there any further clarification or modifications we could make to improve your score?

---

> > ### Comment · Reviewer_pVWC · 2025-04-04
> >
> > Thank you for the reply and for considering my requests.
> >
> > >  the goal of our work is to attempt to evaluate how “good” the basis of features that SAEs discover is by studying how helpful they are on the downstream task of probing. This question is both more useful for practitioners and easier to answer than pure interpretability metrics. Thus, our work also provides a more objective (if indirect) measure of SAE latent interpretability.
> >
> > Yes, I understood this point, but I am not convinced this is a "more objective, indirect measure of interpretability".
> > My concerns were more about Sec. 4.2 and as I already expressed my concerns, the results do not complete the picture on SAE interpretability.

---

### Official Review · Reviewer_gmH6 · 2025-03-14

**Overall Recommendation:** 2

**Summary:**

The paper comprises of several experiments evaluating the efficacy of sparse autoencoder (SAE) approaches to probing.

The paper first focuses on the accuracy of probes under various settings (such as imbalanced data), and finds that SAEs do not improve upon baselines. The paper introduces a "quiver of arrows" methodology, in which the SAE approach is evaluated according to its ability to improve upon a "toolkit" of other methods (specifically, the best method is chosen on a validation set, and the test accuracy is computed).

The paper then examines other potential benefits of SAEs beyond accuracy improvements, such as interpretability. Here, the paper demonstrates that many of the tasks that can be achieved by SAEs can also be achieved by simple baselines (like logistic regression).

**Claims And Evidence:**

The paper's primary claim is that SAE-based probing does not outperform simple baselines. This claim (when restricted to the paper's implementation of SAE-based probing) is supported, on one hand, by a large set of experiments. However, the method by which SAE-based probing is evaluated ("quiver of arrows") is nonstandard. The claim that such an approach is needed for robustness is not convincing to me, as it is widely practiced to compare the individual accuracies of different models.

At a higher level, the claim that SAE-based probing does not have advantages for interpretability is limited by the fact that there are not comparisons to baselines for the most standard interpretability task (interpreting latents). The paper claims that "these findings may be possible using baseline classifiers," (line 373), but this is limited to theoretical speculation. An evaluation here would be more convincing. In general, the other evaluations in Section 4 seemed to be somewhat ad hoc / anecdotal.

**Essential References Not Discussed:**

NA

**Experimental Designs Or Analyses:**

See above

**Methods And Evaluation Criteria:**

See above.

**Other Comments Or Suggestions:**

NA

**Other Strengths And Weaknesses:**

The paper focuses on an important task, evaluating SAEs against baselines, and provides evidence (through many datasets) that SAE-based probing may not offer the improvements that some past work has suggested. The breadth of experiments is a strength, exploring the settings that may particularly illustrate the benefits of SAEs. This kind of fair evaluation is important.

The primary weaknesses of the paper are the nonstandard evaluation strategy in Section 3, and the incomplete and somewhat ad hoc results in Section 4 (in particular, the lack of comparison to baselines in Section 4.2, since interpretability is the primary strength of SAEs).

**Questions For Authors:**

NA

**Relation To Broader Scientific Literature:**

The paper contributes to a growing literature studying SAEs as a tool for mechanistic interpretability. While there has been a lot of excitement regarding the potential of SAEs, as the paper notes, the literature evaluating the practicality/usefulness of SAEs is limited. This paper adds to this literature by considering SAE probing performance on a large number of probing datasets.

**Theoretical Claims:**

NA

---

> ### Author Rebuttal · Authors · 2025-03-31
>
> We are thankful for your time and help, especially related to your points about the quiver of arrows and the clarity of our interpretability experiments. We were glad to hear that you appreciated the breadth of our experiments and found the problem we are investigating important.
>
> ---
>
> > However, the method by which SAE-based probing is evaluated ("quiver of arrows") is nonstandard. The claim that such an approach is needed for robustness is not convincing to me, as it is widely practiced to compare the individual accuracies of different models.
>
> Thank you for bringing up this point! The quiver of arrows approach we introduce is non-standard in the literature, but we adopt it to make the strongest possible case for SAEs. Since we select the best method using validation AUC, we expect to choose SAEs only for tasks where they perform best. To verify this, in the three settings where we employ the quiver of arrows—standard conditions, data scarcity, and class imbalance—we compare its performance to that of using a single SAE across all tasks, as shown below.
> | **Setting** | Baseline Quiver | SAEs Quiver | SAEs + Baselines Quiver | LogReg | SAE 16k k=16 | SAE 16k k=128 | SAE 131k k=16 | SAE 131k k=128 | SAE 1m k=16 | SAE 1m k=128 |
> |------------------------|------------------|-------------|---------------------------|--------|---------------|----------------|----------------|------------------|---------------|----------------|
> | **Standard Conditions**| 0.940 | 0.930 | 0.939 | **0.941** | 0.904 | 0.921 | 0.899 | 0.918 | 0.889 | 0.913 |
> | **Data Scarcity** | 0.819 | 0.806 | 0.812 | **0.836** | 0.800 | 0.816 | 0.794 | 0.810 | 0.785 | 0.801 |
> | **Class Imbalance** | 0.921 | 0.906 | 0.916 | **0.929** | 0.898 | 0.909 | 0.890 | 0.906 | 0.882 | 0.899 |
>
>
> Clearly, the quiver of arrows serves as an upper bound on the performance of any individual SAE. As an alternative counterfactual, instead of comparing against a single SAE probe, we select the best SAE for each task using validation AUC (a “quiver of SAEs”) and compare this to the overall quiver of arrows with both baselines and SAEs. Again, the baseline + SAEs quiver outperforms the just SAE quiver.
>
> > At a higher level, the claim that SAE-based probing does not have advantages for interpretability is limited by the fact that there are not comparisons to baselines for the most standard interpretability task (interpreting latents). The paper claims that "these findings may be possible using baseline classifiers," (line 373), but this is limited to theoretical speculation. An evaluation here would be more convincing. In general, the other evaluations in Section 4 seemed to be somewhat ad hoc / anecdotal.
>
> Thank you for pointing out that line 373 was confusing! In section 4.3, we zoom into two datasets as case studies and show that this is not just a theoretical worry: baseline classifiers can find spurious correlations and noisy labels as well as SAEs. We have modified this paragraph to be as follows:
>
> ```
> The spurious latent category seems especially promising because finding a spurious latent may help us identify spurious features in the dataset. However, in a cast study in \cref{sec:ai_vs_human}, we find that similar findings may be possible using baseline classifiers: we apply a logistic regression probe to model hidden states on tokens from the Pile \cite{pile} and show that maximally activating examples also exhibit the spurious correlation.
>
> However, a practical advantage for SAEs is that the infrastructure to perform autointerp is pre-existing through platforms like Neuronpedia, and a theoretical advantage is that the baseline classifier can only identify the single most relevant coarse-grained feature, while the decomposability of SAE probes into latents allows for identifying many independent features of various importance.
> ```
>
> For your first point, we are not sure what you mean by comparisons to baselines for interpreting latents. Because SAEs are an unsupervised method, it is not clear what a baseline for interpreting a latent would be, as other methods do not have an equivalent to latents (i.e. units a probe can be broken down into). This decomposition is an advantage of SAEs, but it is unclear how much value it gives. In section 4.2, we are investigating what we can learn about different SAE latents by examining the datasets that they are discriminative for (as opposed to auto-interp or human interpretability of latents, which typically looks at top activating examples).
>
> ---
>
> Thank you again for taking the time to review the paper and providing helpful feedback! Do the above actions address your concerns with the paper? If not, what further clarification or modifications could we make to improve your score?

---

### Official Review · Reviewer_9uBB · 2025-03-14

**Overall Recommendation:** 4

**Summary:**

In this work, the authors propose a fair evaluation of SAEs by modelling them as a tool in a practitioner's toolkit, or “quiver of arrows” with the overall question of asking “When is it useful for a practitioner to incorporate SAE probes into their downstream application?”

**Claims And Evidence:**

The author’s claims are supported by relatively clear evidence. The claims are simple and easily validated by the provided experiments. While I do observe evidence that SAE probes contribute less to downstream use cases than other probes, the gap is not incredibly large, especially in Fig. 5. I appreciate the authors claim that autointerpretability methods could easily be applied to probes to achieve similar latent interpretability benefits. However, I was unconvinced by the claims in Section 5 (see strengths/weaknesses). I don’t think the section adds a lot to the impact of the paper and I would encourage removing or deferring it to the appendix.

**Essential References Not Discussed:**

I think the work covered most essential literature.

**Experimental Designs Or Analyses:**

I checked the general quiver-of-arrows AUC comparison setup which is used throughout their experiments and found it to be sound.

**Methods And Evaluation Criteria:**

The quiver of arrows methodology makes sense, especially in a setting where one has multiple SAE probes. Additionally the authors choose a comprehensive suite of probing tasks to cover a wide range of potential use cases for SAE probes. I also think their experiments were thorough and together paint a clear picture of their argument and conclusion. However, I would like to see more elaboration of why the imbalanced dataset settings give the SAE’s inductive bias an advantage.

**Other Comments Or Suggestions:**

* In 2.3, when introducing your probing methods, you seem to be listing the hyperparameters for each method. However, they are introduced as sentence fragments and without context. The writing should be improved here.

 * "throughout the paper we train probes using the largest L0 for SAEs width = 16k, width = 131k, and width = 1M. We use k = 16 to construct easily interpretable probes that potentially overfit less and use k = 128 for performance." What is the “largest l0”?

 * I feel like the “Quiver of Arrows” setup is a core part of your model for practitioner usefulness, yet you describe it only in the Appendix. I would appreciate seeing this discussed in the main body.

 * You may consider flipping the x-axis of Figure 19 and relabel it “Number of Pruned Latents”, as I intuitively read the graph as further right meant more pruning.

 * “We find that ~25% of CoLA labels…” There is a typo here with the ~

**Other Strengths And Weaknesses:**

Section 5 is a little confusing to me. Bricken et al. also use max pooling on their baseline probes, and find that the performance is similar. Thus, it is hard to argue they present an “illusion” of SAE probes being better and thus the need for the argument presented in Figure 11. Additionally, it is hard to understand if your results in the third graph of Figure 11 came from softmax-pooling or the quiver approach. It seems as though you use the quiver approach to “select between pooled and last-token strategies,” does this mean that you are considering all possible strategies {softmax, last} x {SAE, activations} and picking the best? Also, why do the win rates not sum to 100%?

**Questions For Authors:**

Why did you have to run autointerp on your latents, for example in 4.1 and 4.2? Did your GemmaScope and LlamaScope SAEs not come with already-interpreted latents? Also, the autointerp process you used is not described in your paper. There are various strategies to do this proposed by different works so it is important you explain which method you did.

In section 4.2 when you consider the top 128 latents is this again by mean difference across positive and negative samples in your binary classification tasks?

**Relation To Broader Scientific Literature:**

This work ties in well with existing discussion on Sparse Autoencoders in the interpretability literature.

**Theoretical Claims:**

The authors make no theoretical claims in this work.

---

> ### Author Rebuttal · Authors · 2025-03-31
>
> Thank you for your time and help! We were very glad to hear that you appreciated our use of the quiver of arrows technique and found our evidence and claims clear.
>
>
> ---
>
>
> > While I do observe evidence that SAE probes contribute less to downstream use cases than other probes, the gap is not incredibly large, especially in Fig. 5.
>
> Thank you for pointing this out! Figure 5 shows the result of the quiver of arrows approach, and so the SAE quiver achieving the same performance as a baseline quiver only shows that SAEs do not actively make the practitioner worse off. We have added a new table in the appendix which directly compares each SAE to logistic regression.
> | **Setting** | Baseline Quiver | SAEs Quiver | SAEs + Baselines Quiver | LogReg | SAE 16k k=16 | SAE 16k k=128 | SAE 131k k=16 | SAE 131k k=128 | SAE 1m k=16 | SAE 1m k=128 |
> |------------------------|------------------|-------------|---------------------------|--------|---------------|----------------|----------------|------------------|---------------|----------------|
> | **Standard Conditions**| 0.940 | 0.930 | 0.939 | **0.941** | 0.904 | 0.921 | 0.899 | 0.918 | 0.889 | 0.913 |
> | **Data Scarcity** | 0.819 | 0.806 | 0.812 | **0.836** | 0.800 | 0.816 | 0.794 | 0.810 | 0.785 | 0.801 |
> | **Class Imbalance** | 0.921 | 0.906 | 0.916 | **0.929** | 0.898 | 0.909 | 0.890 | 0.906 | 0.882 | 0.899 |
>
> > However, I was unconvinced by the claims in Section 5 (see strengths/weaknesses)...
>
> Thanks! We agree and have moved it to the appendix.
>
> > However, I would like to see more elaboration of why the imbalanced dataset settings give the SAE’s inductive bias an advantage.
>
> Thank you for pointing this out! We have created a new Appendix section to detail the general SAE probing intuition and for each section. We paraphrase below:
>
> ‘’’
> “..We argue that if SAEs are successful at this task [creating a sparse interpretable model basis], requiring a probe to only use a sparse set of directions in this basis should serve as a beneficial inductive bias to prevent overfitting with limited data.”
> Class Imbalance: Because SAE latents are sparsely activating, choosing SAE latents that are positive on the minority class and negative on the majority class may generalize well.
> ‘’’
>
> > Section 5 is a little confusing to me. Bricken et al. also use max pooling on their baseline probes… Thus, it is hard to argue they present an “illusion” of SAE probes being better…
>
>
> Thank you for pointing this out! We attempt to argue that max pooling activations is not the strongest baseline possible. That being said, we agree that our language regarding an “illusion” is too strong, and we have toned down the language in the paper.
>
>
> > Additionally, it is hard to understand if your results in the third graph of Figure 11 came from softmax-pooling or the quiver approach… Also, why do the win rates not sum to 100%?
>
> We apologize that we were not more clear here! The graph compares two quivers, quiver(SAE max pool, SAE last token) vs. quiver(activations softmax,activations last token). The win rates do not sum to 100% because we consider test AUCs within 0.005 to be tied, and thus count as a win for neither method. We have added additional clarification to the text and figure caption.
>
> > In 2.3, when introducing your probing methods, you seem to be listing the hyperparameters for each method…The writing should be improved here.
>
> We agree, thank you! We have now formatted this section as a table.
>
> > …”we train probes using the largest L0”...What is the “largest l0”?
>
> Thank you for noting this! For any given SAE width, we use the largest available L0 from GemmaScope (and noted this in the paper).
>
> > …the “Quiver of Arrows” setup is a core part of your model… I would appreciate seeing this discussed in the main body.
>
> We agree! We moved it to the main body.
>
> > …consider flipping the x-axis of Figure 19…
>
> Thanks, we just made this change!
>
> > …~25%… There is a typo here with the ~
>
> Good catch! We have updated the paper to replace it with the correct value (22%)
>
> > 1)…Did your GemmaScope and LlamaScope SAEs not come with already-interpreted latents? 2
>
> > 2) Also, the autointerp process you used is not described in your paper.
>
> 1)Gemma/LlamaScope SAEs do not come with already-interpreted latents, they just contain the SAEs themselves. 2) we have now added the following language to Section 4.1: “For this and all subsequent experiments, we generate autointerp labels using Neuronpedia, which leverages a language model to produce consistent natural language explanations for a latent based on its top activating tokens.” Thanks for the update!
>
> > In section 4.2 when you consider the top 128 latents is this again by mean difference across positive and negative samples in your binary classification tasks?
>
> Yes, we have fixed this in the text, thank you!
>
> ---
>
>
> Thank you again for taking the time to review the paper and providing helpful feedback! Do the above actions address your concerns with the paper?

---

### Official Review · Reviewer_bCor · 2025-03-23

**Overall Recommendation:** 4

**Summary:**

This paper deals with the problem of evaluating the downstream utility of sparse autoencoders (SAEs). SAEs have recently gained popularity as a means to disentangle concepts learnt by layers of a model, paritcularly LLMs, in order to gain a better mechanistic understanding of their workings. However, evaluating their utility systematically has been challenging. This work examines this by comparing the utility of SAEs with other baseline approaches over a variety of binary tasks. To simulate real world utility, it considers cases of data scarcity, class imbalance, label noise, and covariate shift, and finds that SAE probes do not beat existing baselines. It then performs an analysis over learnt SAE latents to better understand why this happens, and to understand differences in the observations made as compared to prior work.

## Update after rebuttal
Thank you for your response. The most important concerns I had were addressed, so I am increasing my score to accept.

**Claims And Evidence:**

The claims made are generally thoroughly supported by evidence. Some concerns, such as generalization to other layers and datasets, have been raised in the Weaknesses section below.

**Essential References Not Discussed:**

None that I am aware of.

**Experimental Designs Or Analyses:**

Generally the experimental design and analyses appear to be sound and thorough. The paper first explores whether SAEs outperform baselines, and then (in Section 4) looks into possible reasons for not doing so. It also explores why results shown in this work may differ from previous findings (Section 5).

**Methods And Evaluation Criteria:**

The methods and evaluation criteria generally make sense. Some concerns about evaluation have been raised in the weaknesses below, and a critical concern about the method formulation has been raised in Weakness 1 below.

**Other Comments Or Suggestions:**

- L027-033: sentence is hard to read, please rephrase.

**Other Strengths And Weaknesses:**

## Strengths
1. This paper deals with the important problem of assessing the utility of SAEs and to understand if they truly provide an added benefit to existing methods. This has been one of the key promises of SAEs, but has not been evaluated properly so far, and is thus a valuable contribution.
2. The evaluation appears to be very thorough, covering 113 datasets, five baselines, and with several ablations for each.
3. Analyses have also been performed to understand why SAEs may be underperforming (Sections 4 and 5), which can help understand the root cause of the problem and direct future research.

## Weaknesses
1. The input to the SAE probes are the top $K$ latents in the trained SAE that have the maximum absolute difference for the two binary classes (Equation 1). However, since SAE latents (e.g. in TopK SAEs, Gao et al. 2024, L086 right) are not on the same scales, it seems like this could be misleading. For example, consider two latents $A$ and $B$ and classes $0$ and $1$. Let the values of $A$ for hundred data points each of classes 0 and 1 be $\{1,2,\cdots,100\}$ and ${51,52,\cdots,150\}$ respectively. Then the class-wise mean activations would be $50.5$ and $100.5$ respectively, giving a difference (as per Equation 1) of $50$. Now, suppose the values of $B$ for these data points are ${0.01, 0.02, \cdots, 1.00\}$ and $\{10.01, 10.02, \cdots, 11.00\}$ respectively. Then the class-wise mean activations would be $0.505$ and $10.505$ respectively, giving a difference of $10$. Clearly, latent $B$ is more discriminative of the two classes than latent $A$, but the scheme proposed in Equation 1 would pick latent $A$ instead. This could be fixed for instance by normalizing using the mean activations of each latent, and could help avoid misleading conclusions.
2. It is unclear how to interpret Figure 4. If the SAE was "chosen" based on its performance for 14 tasks (L184-188 right), shouldn't the Figure have 14 points above the diagonal? Or is this because the SAE outperformed the baselines in the validation set but underperformed in the test set? If so, the fact that this happened so consistently seems surprising, and a discussion on this would be useful.
3. In Section 4.2 (and Section 4) in general, results from autointerp are assumed to be "ground truths". It would be helpful if this could be evaluated, e.g. using humans for a small subset of latents. As of now it is unclear if the performance loss is due to wrong latents being used (as claimed in Section 4.2) or by autointerp labelling them incorrectly.
4. All evaluation is performed at layer 20 because this is where the baselines performed the best (e.g. as per Section D.1). However, could it be possible that this choice harms the SAEs in the comparison, and that SAEs would perform well at a different layer? A discussion on this would be useful.
5. Results in Section 4, and particularly Section 4.3, are on specific handpicked datasets. Do they generalize? Alternatively, is there a reason for picking these specific datasets? Comment on this would be helpful.
6. Why is a different SAE config used for experiments with label noise (L263) and covariate shift (L294)?
7. In Section 4.1, why is $k=8$ used, when $k=16$ and $k=128$ have been used everywhere else?
8. L158, right: why only use logistic regression for SAE probes?

**Questions For Authors:**

Please refer to the Weaknesses section. Overall, I believe this paper provides a valuable and important contribution, and does a thorough and interesting analysis. I strongly lean towards accept, but I believe the issue with scales of the metric as discussed in Weakness 1 is critical and needs to be addressed. I would be happy to raise my score if this is adequately discussed in the rebuttal.

**Relation To Broader Scientific Literature:**

This work focuses on evaluating the downstream utility of SAEs trained on LLMs. It builds upon recent work (e.g. Bricken et al. 2024, Gao et al. 2024, etc.) that aim to show that SAEs can disentangle concepts learnt by each layer of such models. However, evaluating SAEs for their utility has been limited, for which this work provides an important contribution.

**Theoretical Claims:**

No theoretical claims.

---

> ### Author Rebuttal · Authors · 2025-03-31
>
> Thank you for your insightful questions and comments! We are especially grateful for your suggestions on better latent selection methods, and we are glad you feel our paper is a valuable contribution.
>
> ---
>
> > However, since SAE latents (e.g. in TopK SAEs, Gao et al. 2024, L086 right) are not on the same scales, it seems like this could be misleading.
>
> We agree your method of choosing latents is better, thank you! We find this technique improves test AUC when k is small (<32) but not when k is large (see https://anonymous.4open.science/r/SAE-Probes-B404/rebuttal_plots/comparing_new_and_old_mean_diff_auc.png). Intuitively, your method better finds the “correct” latents for small k, but the old method is “good enough” for large k.
>
> We reran our main experiments with this technique. Unfortunately, the baseline + SAE quiver still fails to improve over baselines. Investigating further, we find that we select k = 128 probes from the quiver 80% of the time, so quiver performance is dominated by k = 128 probes, which do not improve that much.
>
>
> > If the SAE was "chosen" based on its performance for 14 tasks (L184-188 right), shouldn't [Figure 4] have 14 points above the diagonal?
>
> When an SAE is chosen the point may be on or below the diagonal, since the test AUC might decrease or stay about the same. We’ve added “Datasets not directly on the diagonal signify that an SAE method was chosen from the quiver” to the Figure 4 caption.
>
> > In Section 4.2 (and Section 4) in general, results from autointerp are assumed to be "ground truths". It would be helpful if this could be evaluated, e.g. using humans for a small subset of latents.
>
> This is a valid point, thank you! To remove this confounder in Section 4.1, two of the authors independently labeled and ranked latents, with no change in results. However, manually labeling the latents for all datasets in Section 4.2 is labor intensive (and thus a good use case for autointerp).
>
> >could it be possible… that SAEs would perform well at a different layer?
>
> This is a great point that we did not think of, thank you! We checked the width=16k largest L0 SAEs on the 4 layers in Figure 12a (layers 9, 20, 31, and 41), and found that layer 20 was also the best for SAEs as well, see https://anonymous.4open.science/r/SAE-Probes-B404/rebuttal_plots/comparing_sae_test_auc_by_layer.png. We have added this plot and a discussion to Appendix D.
>
>
> > Results in Section 4, and particularly Section 4.3, are on specific handpicked datasets. Do they generalize? Alternatively, is there a reason for picking these specific datasets?
>
> We apologize for not being clearer here; we intended for Section 4.3 to be a set of case studies complementing Section 4.2. We selected these datasets after an investigation of five datasets whose top latent representations exhibited strong performance. During this analysis, we discovered that both 87\_glue\_cola and 110\_aimade\_humangpt3 contained labeling errors. We have altered the introduction to section 4.3 to reflect this.
>
>
> > Why is a different SAE config used for experiments with label noise (L263) and covariate shift (L294)?
>
> For label noise, we use the SAE with smallest width (16k) and maximal L0, which we found to be most performant in standard conditions (we have added this justification to the paper). We would have used this SAE for the covariate shift domain as well, but there is no support for generating autointerp explanations through Neuronpedia for this SAE, so we instead use the width = 131k, L0 = 114 SAE. We have added additional clarification around this choice in the paper, thank you for this comment!
>
> > In Section 4.1, why is k=8 used, when k=16 and k=128 have been used everywhere else?
>
> We do so because the probe pruning experiment is a proof-of-concept experiment and is significantly simpler with smaller k.
>
> > L158, right: why only use logistic regression for SAE probes?
>
> This is a great question! We think this is a good choice because logistic regression is common in practice and is the best baseline activation methods. This is still a valid concern, so we have added the following to our limitations section: "Finally, it is possible that further optimization of the SAE probe baseline might increase performance such that it beats baseline methods. For example, we only tried logistic regression on SAE probes, and it is possible that other probing techniques could perform better."
>
> > L027-033: sentence is hard to read
>
> We agree and have rephrased to: “However, although SAEs occasionally perform better than baselines on individual datasets, we are unable to ensemble SAEs and baselines to consistently improve over just baseline methods.”
>
> ---
> Thank you again for taking the time to review the paper and providing helpful feedback! Do the above actions address your concerns with the paper, especially with regard to the better top k latent selection? If not, what further clarification or modifications could we make to improve your score?

---

### Decision · Program_Chairs · 2025-05-01

**Decision:**

Accept (poster)

**Comment:**

The paper presents a fairly exhaustive systematic empirical evaluation to understand the effectiveness of sparse auto‑encoders (SAEs) in downstream probing of LLM activations. Across several binary classification datasets and different regimes (data scarcity, class imbalance, label noise, covariate shift) the authors compare different SAE variants against simpler baselines on raw activations via an evaluation methodology they refer to as quiver of arrows — the goal is to understand if adding SAE probes to a set of existing baselines improves the performance. The evaluation reveals that performance is not better (and often worse) than the baselines.  Some case studies are used to show that the supposed secondary benefits of SAEs (e.g., detecting spurious correlations, flagging bad labels) can also be matched by simple baselines as well.  These findings urge the community to set a higher empirical bar before claiming interpretability wins from current SAEs.

Given the clear majority of reviewer support, the careful additional experiments, and the paper’s potential impact on a rapidly moving area, I recommend acceptance. For the final version I encourage the authors to include the additional experiments and make the  changes promised in the rebuttal.